# Vestibular Drop Attack: An Analysis of the Therapeutic Response

**Sergio Carmona [1,*], Martin Gabriel Fernandez [1,*] and Cristian David Espona [2]**

1   Fundación San Lucas Para la Neurociencia, Rosario 2000, Argentina
2   Department of Neurology, Hospital Provincial del Centenario, Rosario 2000, Argentina;
    cristianespona@gmail.com
*   Correspondence: sergiocarmona57@gmail.com (S.C.); fmartingabriel@gmail.com (M.G.F.)

**Abstract:** The present study evaluates the response to betahistine in patients who presented vestibular drops attacks in the context of Ménière's disease (MD) and the factors that can predict an unfavorable response to it. A total of 43 patients were analyzed, out of which 33 were diagnosed with MD. This is a descriptive, cross-sectional study with retrospective data collection. Data as regards age, accompanying symptoms, etiological diagnosis and response to MD treatment were collected. A statistical analysis was carried out, and we found that the disease evolution time and specific alterations in the vestibulospinal and oculomotor physical examination present an unfavorable response to betahistine. Failures for betahistine were treated with intratympanic gentamicin, with which symptomatic control was achieved in all cases.

**Keywords:** vestibular drop attacks; Tumarkin attacks; Ménière's disease

## 1. Introduction

Drop attack (DA) is defined as the sudden loss of postural tone that generally causes falls, and which, unlike epilepsy or syncope [1], is not associated with loss of consciousness [2–6]. Its etiology can be explained by vertebrobasilar insufficiency, epilepsy or vestibular instability (the latter are also known as Tumarkin's otolithic crises) [2,3,6–8], and their physiopathology includes the sudden imbalance of the vestibulospinal reflexes induced, for instance, by pathologies such as Ménière's disease (MD), in which fluctuations in vestibular tone are typical [4].

Drop attacks due to a compromise in posterior circulation constitute one of the most difficult differential diagnoses; transient ischemia of bulbar pyramids causes these falls without loss of consciousness, typically with the squat of both legs. Questioning can inquire about the presence of other episodic symptoms during the fall or separated from it, such as blurred vision, double vision, hemianopsia or quadrantanopsia, alternate hemiparesis or hemiparesthesias, and migraine. A vascular history is a key factor.

Tumarkin otolitic crises as a complication of Ménière's disease were first described in 1936 [9], and even today, it is difficult to establish their true incidence. Although new studies would be necessary in order to establish it, a recently published meta-analysis shows a 3 to 19% DA frequency in patients diagnosed with MD, although this varies greatly according to the criteria used for its diagnosis [5,10].

We know that MD is a condition of the inner ear that is characterized by episodic vertigo, auditory fluctuations and ear fullness. Its epidemiology is better known, and it affects approximately 50–200 people per 100,000 inhabitants, and it predominates in patients between the ages of 40 and 60. Its pathophysiology can be explained by the fluctuating endolymphatic hydrops that characterizes it, which is an ideal terrain for the development of sudden pressure changes in the inner ear [3,4,7,11,12]. Although the pathophysiological mechanism that produces DAs is not yet completely clear, it is believed that it could be due to a sudden change in endolymphatic flow, which may result in inadequate otolithic

stimulation [3,7]. Patients describe feeling like they were pushed to the ground or like the surroundings moved [2]. This is what causes the fall, which sometimes even leads to serious injuries [3,13]. This significantly affects the individual's quality of life, and this makes an early diagnosis essential in order to provide a timely treatment. The neurologist should always suspect and be prepared to consider a DA of vestibular origin as a differential diagnosis of the loss of postural tone or instability.

For the treatment of DAs (as well as for the control of MD) we have hygienic–dietary measures, such as salt restriction and pharmacological measures, such as betahistine (BTH) and vestibular sedatives [3], and, if these strategies fail, intratympanic injections gentamicin or steroids can be administered [3,14,15]. We present a series of 43 patients treated at our institution with a diagnosis of DA, of which 33 presented a diagnosis of MD. The purpose of this study is to analyze the characteristics of the treated patients and their response to the established treatment in order to find predictors of adequate or inadequate therapeutic response, which may be useful at the moment of deciding when to escalate the treatment.

## 2. Materials and Methods

This study was approved by the Ad Hoc Ethics Committee of the San Lucas Foundation. A cross-sectional study was conducted; patient data were collected by reviewing medical records between 2015 and 2022 in the database of the "Fundación San Lucas para la Neurociencia", a referral neuro-otological center located in Rosario, Argentina. Patients who presented DA were recorded, and those of vestibular origin were differentiated, especially in the context of Ménière's disease; these were treated with BTH 24 mg twice daily. It is worth mentioning that the patients' personal data were coded in such a way that only researchers can access them. The diagnosis of MD was made according to the diagnostic criteria jointly developed by the Committee for the Classification of Vestibular Disorders of the Bárány Society, the Japan Society for Equilibrium Research, the European Academy of Otology and Neurotology (EAONO), the Balance Committee from the American Academy of Otolaryngology-Head and Neck Surgery (AAO-HNS) and the Korean Balance Society [10]. Although there are different degrees of DA depending on the degree of imbalance suffered by the patient, all the patients included in the series presented severe DA, that is, it caused the patient to fall. Demographic data, such as age and sex, and symptoms such as hearing loss, vertigo, headache, unsteadiness and pulsation, in addition to the vestibular physical examination of the patient (a complete clinical evaluation of the vestibulospinal or oculomotor reflexes), were carried out by experienced neuro-otologists. The time of disease evolution (time elapsed since the appearance of the first symptoms compatible with Ménière's disease, such as vertigo or hearing symptoms) and the response to BTH were recorded in all patients, all of whom were followed up for at least 1 year. Patients who had no DAs after 1-year follow-up were considered to have an adequate response to BTH, whereas patients who had DAs despite being treated with standard doses of BTH for at least 3 consecutive months were considered to have therapeutic failure of BTH. A descriptive analysis of the qualitative and quantitative variables was performed. The former are represented in the tables as frequencies and percentages, while the quantitative variables are summarized as means and standard deviations or, in the case of asymmetrical distributions, as medians and interquartile range (IR) (P25–P75).

The Chi-square, Chi-square test with continuity correction or Fisher's test was applied according to application criteria, to compare qualitative variables between two groups (with and without response to BTH). Student's *t*-test was used once the requirements of randomization, independence, normality and equality of variance were validated. If the normality requirement was not met, the Mann–Whitney U test was used. When significant differences were detected, 95% confidence intervals (CIs) were determined. Initially, a univariate logistic regression analysis was performed, and subsequently, a multivariate logistic regression model was performed to express the strength of the association between the pharmacological response and the variables studied. The OR (odds ratio) and the

CI were estimated at 95%. In all hypothesis contrasts, a significance level of *p* < 0.05 was considered.

Statistical analysis was performed using SPSS 22.0 software (IBM Corporation, New York, NY, USA).

### 3. Results

In a thorough review of the database of patients treated at the San Lucas Foundation between 2015 and 2022, 43 patients with a diagnosis of drop attack were found. All of them underwent neuroimaging (MRI), EKG and cardiologic exams to rule out any other cause that is not MD, pursuant to the diagnostic criteria published by Bárány Society. Ten were excluded, as they did not meet the inclusion criteria for this study. (Nine presented drop attacks of idiopathic etiology, and one patient had not adequately completed the treatment with betahistine.) Therefore, the study population was made up of 33 patients who presented drop attacks of vestibular origin with a diagnosis of Ménière's disease, all of whom were treated with betahistine, and a clinical follow-up of at least 1 year was performed. As shown in Figure 1, 15 patients had an adequate response to BTH; in this group, 6 of them had less than 1 year of duration of MD, while 4 of them had more than 3 years of duration. As for the 18 patients who did not respond to BTH, it should be noted that none of them had less than 1 year of duration, and almost the whole group (17 patients) presented disease that had progressed for more than 3 years. It is important to point out that, in this last group, and given the lack of response to BTH, 12 patients were treated with intratympanic gentamicin (we use doses of 30 mg), with a success rate of 100% with one or several applications; these patients received a clinical follow-up of at least 2 years.

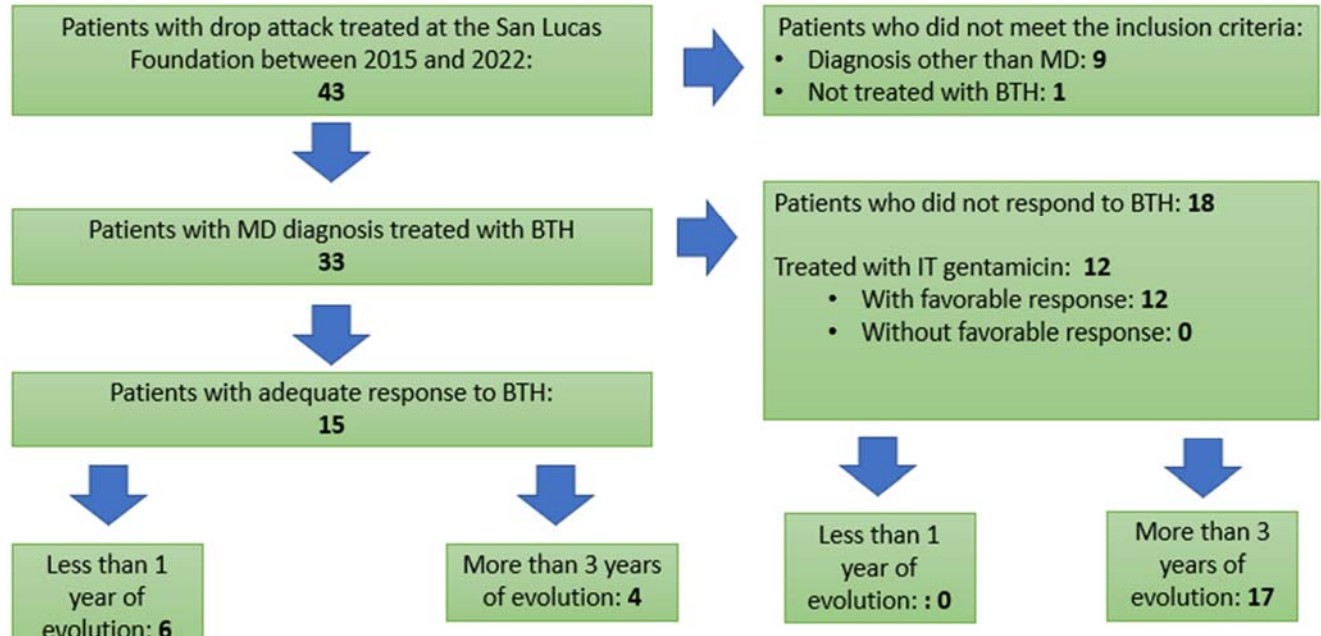

**Figure 1.** Patient's flow chart.

The general characteristics of the 33 patients, as well as their clinical symptoms and comorbidities associated with the drop attack, and the disease evolution time are shown in Table 1.

**Table 1.** General characteristics of the 33 patients.

| Characteristics | N = 33 |
|---|---|
| Average Age (±SD) | 62 (±16) |
| Gender, Male n° (%) | 15 (45) |
| Hearing Loss (%) | 27 (82) |
| Vertigo (%) | 27 (82) |
| Instability (%) | 4 (12) |
| Pulsion (%) | 10 (30) |
| Headache (%) | 6 (18) |
| Abnormal Physical Examination (%) | 24 (73) |
| Active Disease (%) | 25 (76) |
| Migraine (%) | 5 (15) |
| Duration of Symptoms: | |
| Less than 1 year (%) | 6 (18) |
| More than 3 years (%) | 21 (64) |
| Response to betahistine (%) | 15 (45) |

Table 2 shows the differences in the different variables among those patients who showed an adequate response to treatment with BTH and those who did not. Significant differences were found in those patients who presented alterations in the physical examination; here we are referring to patients with alterations in the vestibulospinal or oculomotor (the most frequent clinical findings were nystagmus or positive head impulse tests, as well as deviation of the Bárány indices) in intercritical periods, 53% in the group with a response to BTH as compared to 89% in the group without a response to BTH ($p = 0.047$). On the other hand, we found that in the group of patients who presented a good response to BTH, 40% of them had less than 1 year of symptom duration, while none of the patients in the group who did not respond to BTH had less than 1 year of symptom duration. Likewise, in relation to the evolution time of the disease, we found that, with good statistical significance ($p \leq 0.001$), only 27% of the patients who responded to BTH had a more than 3-year evolution of the disease, while in the group of patients who did not respond to BTH, 94% had a more than 3-year evolution of the disease.

**Table 2.** Distribution of variables according to response to betahistine.

| Variables | With Response (*n* = 15) | Without Response (*n* = 18) | *p* |
|---|---|---|---|
| Age, median (IR) | 65 (56; 73.5) | 63.5 (48; 73) | 0.971 |
| Sex, M n° (%) | 5 (33) | 10 (56) | 0.296 |
| Hearing Loss (%) | 10 (67) | 17 (94) | 0.070 |
| Vertigo (%) | 11 (73) | 16 (89) | 0.375 |
| Instability (%) | 1 (7) | 3 (17) | 0.607 |
| Pulsion (%) | 7 (47) | 3 (17) | 0.125 |
| Headache (%) | 2 (13) | 4 (22) | 0.665 |
| Abnormal Physical Examination (%) | 8 (53) | 16 (89) | 0.047 |
| Active Disease (%) | 11 (73) | 14 (78) | 1.000 |
| Associated Migraine (%) | 2 (13) | 3 (17) | 1.000 |
| Duration of Symptoms: | | | |
| Less than 1 year (%) | 6 (40) | 0 (0) | 0.005 |
| More than 3 years (%) | 4 (27) | 17 (94.4) | <0.001 |

Table 3 shows the results of the univariate logistic regression in relation to the response to BTH in the patients studied. In this model, it is observed, with good statistical power, that the presence of an abnormal physical examination predicts a decrease in the efficacy of BTH (OR: 7.000; CI: 1.173–41.759; $p = 0.033$). Likewise, the history of disease with more than 3 years of duration strongly predicts a failure of BTH for the treatment of drop attacks due to MD according to this model (OR: 46.750; CI: 4.600–475.161; $p = 0.001$).

**Table 3.** Results of the univariate logistic regression analysis to establish the predictive capacity of the different variables in relation to the lack of response to betahistine.

| Variables | Response to Betahistine | | *p* |
|---|---|---|---|
| | **OR** | **CI** | |
| Age | 1.000 | 0.959–1.044 | 0.983 |
| Sex | 2.500 | 0.604–10.440 | 0.206 |
| Hearing Loss | 8.500 | 0.865–83.493 | 0.066 |
| Vertigo | 2.909 | 0.452–18.742 | 0.261 |
| Instability | 2.800 | 0.260–30.178 | 0.396 |
| Pulsion | 0.229 | 0.046–1.340 | 0.071 |
| Headache | 1.857 | 0.290–11.902 | 0.514 |
| Abnormal Physical Examination | 7.000 | 1.173–41.759 | 0.033 |
| Active Disease | 1.273 | 0.258–6.273 | 0.767 |
| Associated Migraine | 1.300 | 0.187–9.021 | 0.791 |
| More than 3 years of duration | 46.750 | 4.600–475.161 | 0.001 |

A multivariate logistic regression analysis was also performed between those variables that showed statistical significance in the univariate model. It is worth mentioning that only the duration of the disease for more than 3 years maintained statistical power in this analysis, with OR: 31.682; CI: 2.350–427.066; *p* = 0.009.

## 4. Discussion

Regarding the general characteristics of the population, the average age is somewhat higher as compared to that described for patients with MD (40 to 60 years), although we can find that it can occur between the third and the seventh decade of life [10]. As a hypothesis, DA would be more frequent in those patients who start presenting symptoms at an older age; the articles that address this topic show ages similar to those found in our study [7]. However, and although this assertion is beyond the objectives of the present study, it is noteworthy that it opens the doors for future research in this regard.

As regards distribution by sex, and in agreement with the literature, a subtle prevalence is observed in the female sex [10]. Overall, 82% of the sample presented the two typical symptoms of the disease, i.e., hearing loss and episodic vertigo, consistent with the literature. Further, 12% of the patients presented instability, and more than 30% of them presented pulsion, symptoms which, although they are not part of the diagnostic criteria, are grounds to suspect the disease [10].

It should be noted that a large proportion of the sample (72%) presented an abnormal vestibulospinal or oculomotor physical examination, although it is fair to note that the examination was carried out by highly trained neuro-otologists, given that San Lucas Foundation is a center of reference on this matter, and this may not be the case in evaluations performed in emergency rooms or general practitioners' offices.

On the other hand, 76% of the patients in the sample had active MD, that is, they had had one or more episodes of vertigo or fluctuating hearing loss in the previous month. As regards the frequency of migraine, we must say that our sample presents an incidence similar to that of the general population, although it is known that the frequency of this disease is higher in patients suffering from MD [16].

It was possible to make an early diagnosis of MD, that is, before one year of symptoms evolution in 18% of the patients; in an identical percentage, diagnosis was made between the first and third year of evolution time of the disease, and, in 64% of the cases, the disease presented over a three-year duration, and, in general, they were patients who had had multiple previous consultations. This situation shows us the importance of maintaining a high level of suspicion in the face of symptoms that are so frequent in general medical and specialized neurological consultations, as is the case with episodic vertigo.

This study was designed to evaluate factors that may predict a poor initial response to BTH in patients treated for drop attacks of vestibular origin secondary to Ménière's

disease. When the variables are individually analyzed, it can be seen that the presence of an abnormal vestibular physical examination or the history of disease with more than 3 years of duration is associated with a poor response to treatment with BTH. We can also say that these two variables show predictive capacity for the failure of BTH as therapy according to the univariate logistic regression model, although it is fair to say that, when analyzing the variables as a whole, only disease of more than 3 years of duration predicts a poor therapeutic response to BTH for the treatment of drop attacks secondary to Ménière's disease [17]. This finding may be due to the fact that the passage of time could produce structural changes in the inner ear that make it less responsive to BTH, although we cannot rule out that we are simply facing patients with more aggressive disease; the findings of this study make it possible for us to support this observation, increasing the cohort would be useful to clear this doubt.

Although in this study we focused on the response to DA treatment in MD and have been able to demonstrate at least a differential response when drug treatment is early versus late and when there is a normal versus abnormal physical examination, we suspect that these conclusions could be extended to patients with MD with a similar severity of rotatory attacks, but without DA. However, it would be necessary to perform a similar experiment involving patients with these characteristics to prove it.

It should be noted that no significant differences were found in the variables sex and age, nor for accompanying symptoms, such as hearing loss, vertigo, pulsion, instability or headache; the history of associated migraine was not related to the therapeutic response to BTH either.

In specialized neuro-otology consultations due to drop attack, we observed that 79% of them were of vestibular origin and had a diagnosis of Ménière's disease. Of them, 39% showed a favorable response to betahistine. We have found that those patients who have alterations in the vestibulospinal or oculomotor physical examination in intercritical periods showed an unfavorable response to the control of the disease with betahistine, as well as those who have 3 years or more of symptoms duration.

An interesting fact of the present series of patients is the high response to treatment with intratympanic gentamicin in those patients who did not respond to BTH [18–20]. A noteworthy fact is that DA control was achieved in 100% of the patients treated with IT gentamicin, either with one or more applications (a maximum of three IT gentamicin applications were performed) [18–20].

Although in the present study we have reviewed the literature analyzing the response to IT gentamicin in patients with MD, we have found few studies analyzing the response to this treatment in patients presenting with DA secondary to MD [21,22]. According to the literature, IT gentamicin controls MD vestibular symptoms in 90–100% of the cases with either one or more applications, which coincides with what we found in our experience in relation to DA.

## 5. Conclusions

Taking into account that DAs are one of the most severe manifestations of MD, and that they have a great impact on the patient's quality of life and can cause serious injuries and even an incapacity to work [23], it is of vital importance to find predictors of therapeutic response that will help us find a way to control this dangerous symptomatology more quickly.

The present study supports our ability to state that in patients who present severe DA secondary to MD, the response to pharmacological treatment with BTH is conditioned, on the one hand, by the duration of symptoms, with the efficacy very good in patients with less than a year of MD duration and poor when MD has been present for more than three years. On the other hand, another strong predictive factor of a bad response to BTH is the presence of an impaired vestibulospinal or oculomotor physical examination in intercritical periods, although, to be rigorous, we may only sustain this last statement when it is carried out by people with expertise in this matter.

The conclusions we were able to obtain open the door to different questions. Bearing in mind that we have found strong predictors of a poor response to BTH, such as an abnormal vestibular physical examination and disease with an evolution time of more than 3 years, is it possible to directly consider treatment with intratympanic gentamicin in those patients who present all these characteristics? In case a treatment with BTH is attempted in these patients, for how long do we apply it before considering it a therapeutic failure? [17,18].

It would also be necessary to carry out a similar experience with an exhaustive statistical analysis in all the patients with MD in order to determine if the relationships found at the time of controlling the vestibular DA have the same implications for the control of the symptoms and the evolution of the disease, which would expand the number of patients who would possibly benefit from these findings.

The main limitation of this study is that it is not a randomized, double-blind, placebo controlled trial, which would be the ideal method to build scientific evidence; but, apart from that, it must be pointed out that this is a retrospective study with a solid statistical method. Also, another limitation we must mention is the small number of participants, due to the fact that we are dealing with a complication of a rare disease, and that in order to avoid diagnostic mistakes, we have only included severe DAs, i.e., those that resulted in a fall to the ground. Finally, we cannot affirm that an early treatment with BTH may guarantee that patients will not present DA episodes, as it is possible that, during the natural course of MD, patients may no longer respond to BTH; for this, a study with a different design would be necessary.

**Author Contributions:** Although all the authors participated in the different sections of the article, some had greater participation in some particular areas. M.G.F. and C.D.E.: Designed the paper, the ladder of probability and the criteria for MD Definitive and Probable; Edited and conducted statistical analysis; Investigated and wrote the part related to Drop Attack based on experience with our own database; Contributed the medical literature. S.C.: Edited the paper, the history of the topic and the contribution of the medical literature; Contributed on the experience in treatment with intratimpanic gentamicin; Contributed to the differential diagnosis between vertebrobasilar TIA and DA; Coordinated the research process and final editing of the manuscript. All authors have read and agreed to the published version of the manuscript.

**Funding:** This research received no external funding.

**Institutional Review Board Statement:** This study was approved by the Ad Hoc Ethics Committee of the San Lucas Foundation (Approval Code: 91218; Approval Date: The present study began on 4 May 2015 and continued until 10 May 2022).

**Informed Consent Statement:** As our study was a retrospective data collection experience, it was not possible to obtain informed consent from the patients. However, in order to protect the confidentiality of the patients who participated, and as explained in the Materials and Methods section, names and surnames were replaced by an alphanumeric code to which ONLY the authors of the study had access, and, for no reason whatsoever, was this code provided to anybody who was not part of the study.

**Data Availability Statement:** Fundación San Lucas declares that the authors can access the data.

**Conflicts of Interest:** The authors declare no conflicts of interest.

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
