# Peer review of "Vestibular Drop Attack: An Analysis of the Therapeutic Response"

_audiolres, doi:10.3390/audiolres14010004_

Round 1

Reviewer 1 Report

Comments and Suggestions for Authors

The paper describes the effect of betahistine on Vestibular Drop Attacks (VDA) in Meniere’s disease (MD). The topic is important as very few papers are focused on this important topic in MD. VDA can led to severe injuries and are impacting the health and work ability of the patients with MD severely. Therefore, the topic is needed for readers to remind of the impact of severe MD.

However, the paper in the present form arises several questions that must be answered before it can be published. The paper is in the present form also too long, especially in the discussion with some repetition. Please shorten and avoid unnecessary statements as “Bearing in mind that the natural  evolution of untreated or insufficiently treated MD is towards hearing loss, it is of vital  importance to have these predictors that can help us carry out timely therapeutic interventions, sparing our patients a long-term disability ” . There is no indication that any treatment will prevent from progression of hearing loss. Please be careful with scientific statement and wishful thinking.

Please use short statements and avoid statements as “First of all, we must say that, in terms of the…”

A medical editor should read the paper and correct the sentences.

The most important remarks are however missing of the referee group in that the patients could be matched with patients with similar severity of the rotatory attacks but without VDA. This would allow comparison of efficacy of betahistine on VDA as I suspect that the efficacy of betahistine is not different between these patient groups. Especially the patients with shorten duration of MD may be responsive to betahistine by “care taking response”.

The onset of MD is also problematic. Does the onset mean start of fist symptom eg. tinnitus in the patient. In less than40 % of the patients all complaints start at the same tine whereas in 20% it takes 5 years or more before hearing loss and vertigo spells occurs in the patients. This should be clarified as in patients with shorten duration of MD the betahistine worked.

The follow up time was not reported. Therefore, relapses with gentamicin cannot be confirmed.

Please instead of “evolution” use the term duration, if you mean duration of the MD

The conclusion is usually quite short. Pease remove “wishful thinking” on this part and shorten it.

Some smaller remarks:

There is no statements of ethical permission.

What were the “the vestibular physical examination of the patient” row 71, p 2

What is BTH row 74 page 2, please define the abbreviation for betahistine

Exclusion criteria row 95, p2 is not complete. One would expect that all cases had passed EKG and CT or MRI

Evolution of the disease MD or DA row 99 p3 is unclear. Please refer to my comments above of duration of MD.

Table 1 from 33 patients there are decimals expressed. If the number of subjects studied is less than 100 decimals should not be used.

Picture 2 shows the differences in the different variables between those patients who 113 showed an adequate response to treatment with BTH and those who did not. Row 113, p3 I do not find any picture 3 in this paper. Do you mean table 2 ?

“On the other hand, response to BTH was found in 40% of the  patients with less than 1 year of evolution of the disease; in contrast, none of the patients in the group that did not respond to BTH had less than 1 year of evolution (p= 0.005).Also, in relation to the disease evolution time, it was found with good statistical significance.” Row 119-125. P 3. The table shows that there are in non-responder group no patient with les than 1 y. duration of MD. Therefore, this statistical evaluation is not relevant.  

Picture 3 shows the results of the univariate logistic regression row 127, p3 No such pictures can be found ? Do you mean table 3 ?

“It was possible to make an early diagnosis of MS, that is, before one year of symptoms evolution in18.2% of the patients; in an identical percentage, diagnosis was made between the first and third year of evolution of the disease, and, in 63.6% of the cases, the disease presented over a three-year evolution, and, in general, they were patients who had made multiple previous consultations.” Row 164-187. P5.  What is MS ? The meaning of the sentence is unclear. Please clarify.

What clinical findings were significant in DA?

The mechanism of Betahistine in DA are discusssed “Although the mechanism of action of BTH is not completely known, we know that it  is a histamine H1 receptor agonist and H3 antagonist, which exerts its beneficial effects on MS at two levels, 1) increasing blood flow at the vestibular level through an improvement in the microcirculation of the stria vascularis in the cochlea, and 2) improving the central compensation of vestibular imbalances through the stimulation of histamine synthesis in the vestibular nuclei. Row 197-201 p 6. The action of betahistine on MD is still unclear as also it’s effectiveness in MD. Please be careful with scientific statement and wishful thinking.

Comments on the Quality of English Language

Please use short statements and avoid statements as “First of all, we must say that, in terms of the…”

A medical editor should read the paper and correct the sentences.

Author Response

Response to Reviewer 1 Comments

1. Summary

2. Questions for General Evaluation

Reviewer’s Evaluation

Response and Revisions

Does the introduction provide sufficient background and include all relevant references?

Must be improved

corresponding response in the point-by-point response letter.

Are all the cited references relevant to the research?

Yes

corresponding response in the point-by-point response letter.

Is the research design appropriate?

Must be improved

corresponding response in the point-by-point response letter.

Are the methods adequately described?

Can be improved

corresponding response in the point-by-point response letter.

Are the results clearly presented?

Can be improved

corresponding response in the point-by-point response letter.

Are the conclusions supported by the results?

Must be improved

corresponding response in the point-by-point response letter.

3. Point-by-point response to Comments and Suggestions for Authors

Comments 1: However, the paper in the present form arises several questions that must be answered before it can be published. The paper is in the present form also too long, especially in the discussion with some repetition. Please shorten and avoid unnecessary statements as “Bearing in mind that the natural evolution of untreated or insufficiently treated MD is towards hearing loss, it is of vital  importance to have these predictors that can help us carry out timely therapeutic interventions, sparing our patients a long-term disability ” . There is no indication that any treatment will prevent from progression of hearing loss. Please be careful with scientific statement and wishful thinking.

Response 1: The statement has been modified to point out that the benefit of finding predictors that may contribute to better management of the disease could result in an improvement in the patient's quality of life. Thanks for pointing this out, we agree with this comment. Therefore, we have modified the wording on the page 2, paragraph 1, and line 53 and we transcribe it below.

“Taking into account that DA is one of the most severe manifestations of MD, and entails a great alteration in the quality of life of the individual, potentially causing serious injuries and even work disability with its economic consequences, it is vitally important to found predictors that can help us to control this dangerous symptomatology more quickly in order to avoid greater harm to the patient.”

Comments 2: Please use short statements and avoid statements as “First of all, we must say that, in terms of the…”

Response 2: We have modified the way it is written to make it more appropriate. Thanks for pointing this out, we agree with this comment. Therefore, we have modified the wording on the page 6, paragraph 1, and line 160 and we transcribe it below

“Regarding the general characteristics of the population, the average age is somewhat higher compared to that described for patients with MD (40 to 60 years), although we can find that it can occur between the third and the seventh decade of life (17).”

Comments 3: The most important remarks are however missing of the referee group in that the patients could be matched with patients with similar severity of the rotatory attacks but without VDA. This would allow comparison of efficacy of betahistine on VDA as I suspect that the efficacy of betahistine is not different between these patient groups. Especially the patients with shorten duration of MD may be responsive to betahistine by “care taking response”

Response 3: We have added a paragraph to the discussion in order to highlight the possible extrapolation of the findings to the population of patients with severe MD, as suggested. Thanks for pointing this out, we agree with this comment. Therefore, we have added this text on the page 7, paragraph 5, and line 222 as transcribed below

“Although in this study we focused on the response to DA treatment in MD and have been able to demonstrate at least a differential response when drug treatment is early versus late and when there is a normal versus abnormal physical examination, we suspect that these conclusions could be extended to patients with MD with a similar severity of rotatory attacks but without DA. However, it would be necessary to perform a similar experiment involving patients with these characteristics to prove it.”

Comments 4: The onset of MD is also problematic. Does the onset mean start of fist symptom eg. tinnitus in the patient. In less than40 % of the patients all complaints start at the same tine whereas in 20% it takes 5 years or more before hearing loss and vertigo spells occurs in the patients. This should be clarified as in patients with shorten duration of MD the betahistine worked.

Response 4: In the materials and methods section, we have added the definition of duration of the disease, to make it easier to understand; we have also modified the wording in relation to this throughout the text to better convey the general idea. Thanks for pointing this out, we agree with this comment. Therefore, we have added this text on the page 2, paragraph 2, and line 75 as transcribed below

“Demographic data such as age and sex, symptoms such as hearing loss, vertigo, headache, instability, pulsion, in addition to the vestibular physical examination (alterations in the vestibulospinal or oculomotor physical examination) of the patient, the most frequent clinical findings were nystagmus or positive head impulse tests, as well as deviation of the Bárány indices. We have also found a considerable group of patients with unilateral hearing loss. disease evolution time (assumed to be the time elapsed since the onset of the first symptoms compatible with Ménière´s disease such as vertigo or auditory symptoms) and response to betahistine were recorded for all patients, those patients who were free of DA after a 1 year follow-up were considered as having an adequate response to BTH, while those patients that presented DA despite being under treatment with maximum doses of BTH (48 mg per day) during more than 3 months were considered a therapeutic failure to BTH.”

Comments 5: The follow up time was not reported. Therefore, relapses with gentamicin cannot be confirmed.

Response 5: We have reviewed the clinical records and clarified the follow-up time of the patients who were treated with IT gentamicin, for better reading comprehension. Thanks for pointing this out. We have added this text on the page 3, paragraph 3, and line 118 as transcribed below

“It is important to point out that, in this last group, and given the lack of response to BTH, 12 patients were treated with intratympanic gentamicin with a more than acceptable success rate of 100% with 1 or several applications; these patients received a clinical follow-up of at least 2 years.”

Comments 6: Please instead of “evolution” use the term duration, if you mean duration of the MD

Response 6: Agreed. Accordingly, we have reviewed the entire text to emphasize this point. Thanks for pointing this out.

Comments 7: The conclusion is usually quite short. Pease remove “wishful thinking” on this part and shorten it.

Response 7: Agreed. Consequently, we have deleted a part of the text that we understand is “wishful thinking” to emphasize this point, and we have shorten the conclusion section, as has been suggested. Thanks for pointing this out. We have deleted the text transcribed below.

“We think that one of the next challenges would be to adapt therapeutic behaviors to these new prognostic factors in order not to unnecessarily waste time, always with the goal of improving our patients’ quality of life.”

Comments 8: There is no statements of ethical permission

Response 8: As this is a retrospective data collection study, the data have been coded so that only the researchers may have access to the private information of the patients involved. Accordingly, this is mentioned in the materials and methods section. Likewise, this study was approved by the corresponding Ethics Committee of the San Lucas Foundation. Thanks for pointing this out, we agree with this comment. Therefore, we have added this text on the page 2, paragraph 2, and line 60 and 66 as transcribed below

“This study was approved by the Ad Hoc Ethics Committee of the San Lucas Foundation.”

“It must be pointed out that the patients’ personal data was codified in such a way that only the researchers may have access to them.”

Comments 9: What were the “the vestibular physical examination of the patient” row 71, p 2

Response 9: Vestibular physical examination refers to the alterations in the vestibulospinal or oculomotor systems which were observed during the neurotologic assessment. Thanks for pointing this out, we agree with this comment. Therefore, we have added this text on the page 2, paragraph 2, and line 75 as transcribed below

“Demographic data such as age and sex, symptoms such as hearing loss, vertigo, headache, instability, pulsion, in addition to the vestibular physical examination (alterations in the vestibulospinal or oculomotor physical examination) of the patient”

Comments 10: What is BTH row 74 page 2, please define the abbreviation for betahistine

Response 10: Agreed. Accordingly, we have reviewed the entire text to emphasize this point. Thanks for pointing this out. We have added this text on the page, paragraph, and line.

“betahistine (BTH)”

Comments 11: Exclusion criteria row 95, p2 is not complete. One would expect that all cases had passed EKG and CT or MRI

Response 11: Agreed. Consequently, we have added the study methodology used to rule out the different differential diagnoses, so as to highlight this point. Therefore, we have added this text on the page 3, paragraph 3, and line 106 as transcribed below:

“All of them underwent neuroimaging (MRI), EKG and cardiologic exams, so as to rule out any other cause that is not MD, pursuant to the diagnostic criteria published by Bárány Society.”

Comments 12: Evolution of the disease MD or DA row 99 p3 is unclear. Please refer to my comments above of duration of MD.

Response 12: Throughout the text, when we speak of duration of the disease, we refer to MD. Accordingly, we have reviewed the entire text to emphasize this point.

Comments 13: Table 1 from 33 patients there are decimals expressed. If the number of subjects studied is less than 100 decimals should not be used.

Response 13: Thanks for pointing this out. Agreed. Therefore, we have made the suggested corrections to Table 1.

 Comments 14: Picture 2 shows the differences in the different variables between those patients who 113 showed an adequate response to treatment with BTH and those who did not.”  Row 113, p3 I do not find any picture 3 in this paper. Do you mean table 2 ?

Response 14: Agreed. Consequently, we have modified the text to stress this point. Thanks for pointing this out, we agree with this comment. Therefore, we have added this text on the page 4, paragraph 1, and line 129 as transcribed below

“Table 2 shows the differences in the different variables between those patients who showed an adequate response to treatment with BTH and those who did not.”

Comments 15: “On the other hand, response to BTH was found in 40% of the  patients with less than 1 year of evolution of the disease; in contrast, none of the patients in the group that did not respond to BTH had less than 1 year of evolution (p= 0.005).Also, in relation to the disease evolution time, it was found with good statistical significance.” Row 119-125. P 3. The table shows that there are in non-responder group no patient with les than 1 y. duration of MD. Therefore, this statistical evaluation is not relevant. 

Response 15: Agreed. Consequently, we have modified the text to stress this point. Thanks for pointing this out, we agree with this comment. Therefore, we have added this text on the page 4, paragraph 1, and line 136 as transcribed below

“On the other hand, response to BTH was found in 40% of the patients with less than 1 year of duration of the disease; in contrast, none of the patients in the group that did not respond to BTH had less than 1 year of onset of symptoms; although we cannot currently establish an adequate statistical association, as we do not have patients who do not respond to BTH with less than 1 year of duration of MD, this situation calls our attention.”

Comments 16: Picture 3 shows the results of the univariate logistic regression row 127, p3 No such pictures can be found ? Do you mean table 3 ?

Response 16: Agreed. Consequently, we have modified the text to stress this point. Thanks for pointing this out, we agree with this comment. Therefore, we have added this text on the page 4, paragraph 2, and line 146 as transcribed below

“Table 3 shows the results of the univariate logistic regression in relation to the lack of response to BTH in the patients studied”

Comentarios 17 : “Se logró realizar un diagnóstico temprano de EM, es decir, antes del año de evolución de los síntomas en el 18,2% de los pacientes; en idéntico porcentaje el diagnóstico se realizó entre el primer y tercer año de evolución de la enfermedad, y en el 63,6% de los casos la enfermedad se presentó en tres años de evolución, y en general eran pacientes que habían realizado múltiples consultas previas ”. Fila 164-187. P5. ¿Qué es la EM? El significado de la frase no está claro. Por favor aclare.

Respuesta 17: Parece ser un error tipográfico, nos referíamos a MD. En consecuencia, hemos modificado el texto para enfatizar este punto. Gracias por señalarlo, estamos de acuerdo con este comentario. Por ello, hemos añadido el texto de la página 6, párrafo 4, línea 200 que transcribimos a continuación.

“Se logró realizar un diagnóstico temprano de EM, es decir, antes del año de evolución de los síntomas en el 18,2% de los pacientes; en idéntico porcentaje el diagnóstico se realizó entre el primer y tercer año de evolución de la enfermedad, y en el 63,6% de los casos la enfermedad se presentó durante tres años y, en general, fueron pacientes que tuvieron realizó múltiples consultas previas”.

Comentarios 18 : Se analiza el mecanismo de la betahistina en la DA. Aunque el mecanismo de acción de la BTH no se conoce completamente, sabemos que es un agonista del receptor H1 de histamina y un antagonista H3, que ejerce sus efectos beneficiosos sobre la EM en dos niveles, 1 ) aumentar el flujo sanguíneo a nivel vestibular mediante una mejora de la microcirculación de la estría vascular en la cóclea, y 2) mejorar la compensación central de los desequilibrios vestibulares mediante la estimulación de la síntesis de histamina en los núcleos vestibulares. Fila 197-201 p 6. La acción de la betahistina en la DM aún no está clara, al igual que su eficacia en la DM. Tenga cuidado con las declaraciones científicas y las ilusiones.

Respuesta 18: De acuerdo. En consecuencia, hemos modificado el texto para enfatizar este punto. Gracias por señalarlo, estamos de acuerdo con este comentario. Por lo tanto, hemos agregado este texto en la página 7, párrafo 7 y línea 223 como se transcribe a continuación.

“Aunque el mecanismo de acción de la BTH no se comprende completamente, se cree que puede ejercer su acción como agonista del receptor H1 de histamina y antagonista H3, lo que ejerce sus efectos beneficiosos en la EM en dos niveles: 1) aumentando el flujo sanguíneo en el nivel vestibular a través de una mejora en la microcirculación de la estría vascular en la cóclea, y 2) mejora de la compensación central de los desequilibrios vestibulares a través de la estimulación de la síntesis de histamina en los núcleos vestibulares”.

Reviewer 2 Report

Comments and Suggestions for Authors

This is an interesting retrospective case series on the response to treatment of patients with Menière’s disease (MD) and drop attacks with betahistine and intratympanic gentamicin. The data suggests that early treatment with betahistine is effective for prevention of drop attacks whereas patients do not benefit from betahistine when treatment is initiated later in the course. For this reason the authors stress the importance of an early diagnosis of MD. However, their data does not clearly show that early treatment with betahistine leads to abolition of drop attacks also later in the course. This issue should be discussed. Likewise, the authors should compare their findings with published studies on betahistine in MD and drop attacks. Have other studies been published that compare early versus late treatment with betahistine? Have the authors also analysed the effect of betahistine on attacks of vertigo in their patients?

Sixteen of 18 patients not responding to betahistine were treated with intratympanic gentamicin with favorable response. I suggest that the authors discuss this finding a bit more in detail and compare it to published studies on intratympanic gentamicin in patients with MD and drop attacks. The methods of intratympanic gentamicin injections should be included in the paper, e.g. dosage. 

Treatment response needs to be clearly defined and the reader wants to know how long the patients have been followed up. Can the authors provide any information on the frequency of drop attacks in their patients before treatment?

Page 1: “Its etiology can be explained by vertebrobasilar insufficiency, epilepsy or vestibular instability….”. Maybe the authors should mention that the etiology of drop attacks is often cryptogenic. 

Page 1: “Although new studies would be necessary in order to establish it, a recently published meta-analysis shows a 3 to 19% DA frequency of in patients diagnosed with MD, 30 although this varies greatly according to the criteria used for its diagnosis (16).“ Reference 16 is a historic paper by Tumarkin. The authors should also include the reference of the meta-analysis. 

Page 1: “Vestibular DA is due to a sudden change in endolymphatic flow…”. This is a hypothesis, whereas “is due to” sounds like a matter of fact. Please put it more conservative. 

Page 2, line 47: Intratympanic injection of steroid should also be mentioned as a treatment option.  

Page 2: “Bearing in mind that the natural evolution of untreated or insufficiently treated MD is towards hearing loss, it is of vital importance to have these predictors that can help us carry out timely therapeutic interventions, sparing our patients a long-term disability.” This sentence is misleading as it suggests that treating MD can avoid development of hearing loss. Unfortunately, there is no effective treatment that prevents hearing loss in MD. 

Page 2, line 67: Please cite the original reference published in Journal of Vestibular Research, not the Spanish translation. 

Page 2, line 73: what do you mean with adequate AD control? The criteria of therapeutic failure should be explained in the methods. 

Page 3, line 101: the term “evolution” needs to be explained. Do the authors mean the time since first occurrence of a drop attack?

Page 3, line 106: What is the definition of the success rate? How long was the follow-up?

Table 1: Only 82% of 33 patients with MD had hearing loss and vertigo. The authors need to explain this as hearing loss and episodic vertigo are prerequisites for the diagnosis of MD according to the criteria of the Bárány Society. 

Page 4, line 113: Table 2, not picture 2. 

Page 4, line 116: Please consider to cite the most common pathologic clinical findings. 

Page 4, line 120: “response to BTH was found in 40% of the patients with less than 1 year of evolution of the disease; in contrast, none of the patients in the group that did not respond to BTH had less than 1 year of evolution (p= 0.005).” This sentence is contradictory. 

Page 5, line 164: Do you mean MD?

The limitations of the study should be discussed. 

Comments on the Quality of English Language

well written, only minor mistakes. 

Author Response

Response to Reviewer 2 Comments

1. Summary

2. Questions for General Evaluation

Reviewer’s Evaluation

Response and Revisions

Does the introduction provide sufficient background and include all relevant references?

Must be improved

corresponding response in the point-by-point response letter.

Are all the cited references relevant to the research?

Yes

corresponding response in the point-by-point response letter.

Is the research design appropriate?

Must be improved

corresponding response in the point-by-point response letter.

Are the methods adequately described?

Can be improved

corresponding response in the point-by-point response letter.

Are the results clearly presented?

Can be improved

corresponding response in the point-by-point response letter.

Are the conclusions supported by the results?

Must be improved

corresponding response in the point-by-point response letter.

3. Point-by-point response to Comments and Suggestions for Authors

Comments 1: This is an interesting retrospective case series on the response to treatment of patients with Menière’s disease (MD) and drop attacks with betahistine and intratympanic gentamicin. The data suggests that early treatment with betahistine is effective for prevention of drop attacks whereas patients do not benefit from betahistine when treatment is initiated later in the course. For this reason the authors stress the importance of an early diagnosis of MD. However, their data does not clearly show that early treatment with betahistine leads to abolition of drop attacks also later in the course. This issue should be discussed. Likewise, the authors should compare their findings with published studies on betahistine in MD and drop attacks. Have other studies been published that compare early versus late treatment with betahistine? Have the authors also analysed the effect of betahistine on attacks of vertigo in their patients?

Response 1: The purpose of this study was to find factors that may predict a therapeutic failure in patients with DA secondary to MD; symptoms duration was one of the positive results of this study, and, therefore, we stress the importance of an early diagnosis of this condition; however, we cannot assert that early diagnosis is associated with long-term abolition of DA, as this would require a study with a different design, and this is beyond the objectives of the present study. In any case, it would be interesting to carry out long-term follow-up of this cohort of patients. Besides, we have in the literature not found similar experiences that compare early versus late treatment with betahistine in DA, and that is one of the novel aspects of this experience. Finally, and as we mention in the conclusions section, our future project is to extend the experience to patients with MD in relation to vertigo attacks, so as to expand the number of patients who may potentially benefit from the conclusions. Thanks for pointing this out, we agree with this comment. Therefore, we have added this text on the page 7, paragraph 6, and line 287 as transcribed below:

“it must be pointed out that we cannot assert that an early treatment is associated with long-term abolition of DA, as this would require a study with a different design, and this is beyond the objectives of the present study. In any case, it would be interesting to carry out long-term follow-up of this cohort of patients.”

Comments 2: Sixteen of 18 patients not responding to betahistine were treated with intratympanic gentamicin with favorable response. I suggest that the authors discuss this finding a bit more in detail and compare it to published studies on intratympanic gentamicin in patients with MD and drop attacks. The methods of intratympanic gentamicin injections should be included in the paper, e.g. dosage. Thanks for pointing this out, we agree with this comment. Therefore, we have added this text on the page 7, paragraph 3, and line 256 as transcribed below:

Response 2: Although in the present study we have reviewed the literature that analyzes the response to IT gentamicin in patients with MD, we have not found studies that analyze the answer to this treatment in patients that present DA secondary to MD. According to the literature, IT gentamicin controls MD vestibular symptoms 90-100% of the cases with either one or more applications, which coincides with what we found in our experience in relation to DA.

Although in the present study we have reviewed the literature that analyzes the response to IT gentamicin in patients with MD, we have not found studies that analyze the answer to this treatment in patients that present DA secondary to MD. According to the literature, IT gentamicin controls MD vestibular symptoms 90-100% of the cases with either one or more applications, which coincides with what we found in our experience in relation to DA.

“It is important to point out that, in this last group, and given the lack of response to BTH, 12 patients were treated with intratympanic gentamicin (we use doses of 40 mg) with a more than acceptable success rate of 100% with 1 or several applications; these patients received a clinical follow-up of at least 2 years.” Thanks for pointing this out, we agree with this comment. Therefore, we have added this text on the page 3, paragraph 3, and line 119 as transcribed below

Comments 3: Treatment response needs to be clearly defined and the reader wants to know how long the patients have been followed up. Can the authors provide any information on the frequency of drop attacks in their patients before treatment?

Response 3: We have considered that a good response to betahistanine treatment is seen in patients who are free of DA episodes during at least 1 year. On the other hand, we have considered therapeutic failure those cases that, despite being treated with 48 mg per day in 2 doses of betahistine, continued to have DA after a period of 3 months. Thanks for pointing this out, we agree with this comment. Therefore, we have added this text on the page 2, paragraph 2, and line 83 as transcribed below

“those patients who were free of DA after a 1 year follow-up were considered as having an adequate response to BTH, while those patients that presented DA despite being under treatment with maximum doses of BTH (48 mg per day) during more than 3 months were considered a therapeutic failure to BTH.”

Comments 4: Page 1: “Its etiology can be explained by vertebrobasilar insufficiency, epilepsy or vestibular instability….”. Maybe the authors should mention that the etiology of drop attacks is often cryptogenic.

Response 4: In our case series, DA associated to MS largely predominates, although also to vestibular migraine, BPPV, central vestibular syndromes and those in which we do not have a diagnosis that justifies them. These cases may be considered as cryptogenic. . Thanks for pointing this out, we agree with this comment. Therefore, we have added this text on the page 3 , paragraph 3 , and line 110 as transcribed below:

“10 of which were excluded as they did not meet the inclusion criteria for this study. (9 presented drop attacks of idiopathic etiology and 1 patient had not adequately completed the treatment with betahistine).”

Comments 5: Page 1: “Although new studies would be necessary in order to establish it, a recently published meta-analysis shows a 3 to 19% DA frequency of in patients diagnosed with MD, 30 although this varies greatly according to the criteria used for its diagnosis (16).“ Reference 16 is a historic paper by Tumarkin. The authors should also include the reference of the meta-analysis.

Response 5: We have added a bibliographic reference. Thanks for pointing this out, we agree with this comment. Therefore, we have added this text on the page 1, paragraph 1, and line 29 as transcribed below:

“Although new studies would be necessary in order to establish it, a recently published meta-analysis shows a 3 to 19% DA frequency of in patients diagnosed with MD, although this varies greatly according to the criteria used for its diagnosis (5, 17)”

Comments 6: Page 1: “Vestibular DA is due to a sudden change in endolymphatic flow…”. This is a hypothesis, whereas “is due to” sounds like a matter of fact. Please put it more conservative.

Response 6: We have modified the text to stress this point. Thanks for pointing this out, we agree with this comment. Therefore, we have added this text on the page 1, paragraph 2, and line 37 as transcribed below:

“Although the pathophysiological mechanism that produces DAs is not yet completely clear, it is believed that it could be due to a sudden change in endolymphatic flow, which may result in inadequate otolytic stimulation (1, 4)“

Comments 7: Page 2, line 47: Intratympanic injection of steroid should also be mentioned as a treatment option. 

Response 7: We have modified the text to stress this point. Thanks for pointing this out, we agree with this comment. Therefore, we have added this text on the page 2, paragraph 1, and line 46 as transcribed below

“For the treatment of DA (as well as for the control of MD) we have hygienic-dietary measures such as salt restriction, pharmacological measures such as betahistine and vestibular sedatives (4), and, if these strategies fail, intratympanic injections gentamicin or steroids can be administered (4, 19, 20).”

Comments 8: “Bearing in mind that the natural evolution of untreated or insufficiently treated MD is towards hearing loss, it is of vital importance to have these predictors that can help us carry out timely therapeutic interventions, sparing our patients a long-term disability.” This sentence is misleading as it suggests that treating MD can avoid development of hearing loss. Unfortunately, there is no effective treatment that prevents hearing loss in MD.

Response 8: The statement has been modified to point out that the benefit of finding predictors that may contribute to better management of the disease could result in an improvement in the patient's quality of life. Thanks for pointing this out, we agree with this comment. Therefore, we have modified the wording on the page 2, paragraph 1, and line 53 and we transcribe it below:

“Taking into account that DA is one of the most severe manifestations of MD, and entails a great alteration in the quality of life of the individual, potentially causing serious injuries and even work disability with its economic consequences, it is vitally important to found predictors that can help us to control this dangerous symptomatology more quickly in order to avoid greater harm to the patient.”

Comments 9: Page 2, line 67: Please cite the original reference published in Journal of Vestibular Research, not the Spanish translation.

Response 9: We have modified the way it is written to make it more appropriate. Thanks for pointing this out, we agree with this comment. Therefore, we have modified the wording on the page 2, paragraph 2, and line 69 and we transcribe it below

“José A. Lopez-Escamez a,∗, John Carey b, Won-Ho Chung c, Joel A. Goebel d,Måns Magnussone, Marco Mandalàf, David E. Newman-Toker g, Michael Strupp h, Mamoru Suzukii , Franco Trabalzinif y Alexandre Bisdorffj Diagnostic criteria for Ménière's disease. jointly formulated by the Classification Committee of the Bárány Society, The Japan Society for Equilibrium Research, the European Academy of Otology and Neurotology (EAONO), the Equilibrium Committee of the American Academy of Otolaryngology-Head and Neck Surgery (AAO-HNS) and the Korean Balance Society. Acta Otorrinolaringol Esp. 2015.”

Comments 10: Page 2, line 73: what do you mean with adequate DA control? The criteria of therapeutic failure should be explained in the methods.

Response 10: We have modified the way it is written to make it more appropriate. Thanks for pointing this out, we agree with this comment. Therefore, we have modified the wording on the page 2, paragraph 2, and line 83 and we transcribe it below:

 “those patients who were free of DA after a 1 year follow-up were considered as having an adequate response to BTH, while those patients that presented DA despite being under treatment with maximum doses of BTH (48 mg per day) during more than 3 months were considered a therapeutic failure to BTH.”

Comments 11: Page 3, line 101: the term “evolution” needs to be explained. Do the authors mean the time since first occurrence of a drop attack?

Response 11: In the materials and methods section, we have added the definition of duration of the disease, to make it easier to understand; we have also modified the wording in relation to this throughout the text to better convey the general idea. Thanks for pointing this out, we agree with this comment. Therefore, we have added this text on the page 3, paragraph 3, and line 114 as transcribed below:

“As shown in Picture 1, 15 patients had an adequate response to BTH; in this group, 6 of them had less than 1 year of duration of the MD, while 4 of them had more than 3 years of duration. As for the 18 patients who did not respond to BTH, it should be noted that none of them had less than 1 year of duration, and almost the whole group (17 patients) presented a disease that had progressed for more than 3 years.”

Comments 12: Page 3, line 106: What is the definition of the success rate? How long was the follow-up?

Response 12: We have modified the way it is written to make it more appropriate. Thanks for pointing this out, we agree with this comment. Therefore, we have modified the wording on the page 3 , paragraph 3, and line 111 and we transcribe it below

“Therefore, the study population was made up of 33 patients who presented drop attacks of vestibular origin with a diagnosis of Ménière's disease, all of whom were treated with betahistine,  a clinical follow-up of at least 1 year was performed.”

Comments 13: Table 1: Only 82% of 33 patients with MD had hearing loss and vertigo. The authors need to explain this as hearing loss and episodic vertigo are prerequisites for the diagnosis of MD according to the criteria of the Bárány Society.

Response 13: Ménière's disease is caused by a triad consisting of vertigo episodes, fluctuating hearing and tinnitus. Though most of our patients meet the criteria for definitive MD, some of them presented the clinical triad but this did not translate into hearing loss in audiological studies, or their audiometries presented the typical ascending curve but without vertigo episodes. However, these patients meet the Bárány Society criteria for definitive MD.

Comments 14: Page 4, line 113: Table 2, not picture 2.

Response 14: Agreed. Consequently, we have modified the text to stress this point. Thanks for pointing this out, we agree with this comment. Therefore, we have added this text on the page 4, paragraph 1, and line 129 as transcribed below

“Table 2 shows the differences in the different variables between those patients who showed an adequate response to treatment with BTH and those who did not.”

Comments 15: Page 4, line 116: Please consider to cite the most common pathologic clinical findings.

The most frequent clinical findings were nystagmus or positive head impulse tests, as well as deviation of the Bárány indices. We have also found a considerable group of patients with unilateral hearing loss. Agreed. Consequently, we have modified the text to stress this point. Thanks for pointing this out, we agree with this comment. Therefore, we have added this text on the page 4, paragraph 1, and line 133 as transcribed below:

“the most frequent clinical findings were nystagmus or positive head impulse tests, as well as deviation of the Bárány indices”

Comments 16: Page 4, line 120: “response to BTH was found in 40% of the patients with less than 1 year of evolution of the disease; in contrast, none of the patients in the group that did not respond to BTH had less than 1 year of evolution (p= 0.005).” This sentence is contradictory.

Response 16: Agreed. Consequently, we have modified the text to stress this point. Thanks for pointing this out, we agree with this comment. Therefore, we have added this text on the page 4, paragraph 1, and line 136 as transcribed below:

“On the other hand, response to BTH was found in 40% of the patients with less than 1 year of duration of the disease; in contrast, none of the patients in the group that did not respond to BTH had less than 1 year of onset of symptoms; although we cannot currently establish an adequate statistical association, as we do not have patients who do not respond to BTH with less than 1 year of duration of MD, this situation calls our attention.”

Comments 17: Page 5, line 164: Do you mean MD?

Response 17: It seems to be a typo, we were referring to MD. Consequently, we have modified the text to stress this point. Thanks for pointing this out, we agree with this comment. Therefore, we have added this text on the page 6, paragraph 4, and line 200 as transcribed below:

“It was possible to make an early diagnosis of MD, that is, before one year of symptoms evolution in 18.2% of the patients; in an identical percentage, diagnosis was made between the first and third year of evolution time of the disease, and, in 63.6% of the cases, the disease presented over a three-year duration, and, in general, they were patients who had made multiple previous consultations.”

Comments 18:  The limitations of the study should be discussed.

Response 18: Agreed. Consequently, we have modified the text to stress this point. Thanks for pointing this out, we agree with this comment. Therefore, we have added this text on the page 8, paragraph 1 , and line 292 as transcribed below

“As limitations of the present study we can mention a minor one, which may be corrected in future experiences by including all patients with MS, not only those who suffered from DA, and evaluating their pharmacological response; it must also be pointed out that we cannot assert that an early treatment is associated with long-term abolition of DA, as this would require a study with a different design, and this is beyond the objectives of the present study. In any case, it would be interesting to carry out long-term follow-up of this cohort of patients.”

Reviewer 3 Report

Comments and Suggestions for Authors

This is a very short report including a case series of patients with Meniere's disease starting over 60 years old and with a short duration of the disease. Large series have found that Meniere's usually start at forties, so this series is not representative of Meniere disease.

Did the author investigated for a migraine history? Other comorbidities should also me included such allergy or autoimmune diseases.

How did the authors rule out a transient stroke?

What the vascular risk profile assessed? Cholesterol and high blood pressure may be relevant in these individuals. Please include this information in Table 1.

Author Response

Response to Reviewer 3 Comments

1. Summary

2. Questions for General Evaluation

Reviewer’s Evaluation

Response and Revisions

Does the introduction provide sufficient background and include all relevant references?

Must be improved

corresponding response in the point-by-point response letter.

Are all the cited references relevant to the research?

Must be improved

corresponding response in the point-by-point response letter.

Is the research design appropriate?

Must be improved

corresponding response in the point-by-point response letter.

Are the methods adequately described?

Must be improved

corresponding response in the point-by-point response letter.

Are the results clearly presented?

Can be improved

corresponding response in the point-by-point response letter.

Are the conclusions supported by the results?

Yes

corresponding response in the point-by-point response letter.

3. Point-by-point response to Comments and Suggestions for Authors

Comments 1: This is a very short report including a case series of patients with Meniere's disease starting over 60 years old and with a short duration of the disease. Large series have found that Meniere's usually start at forties, so this series is not representative of Meniere disease.

Response 1: Although the literature consulted establishes that the highest incidence of MD occurs between the 4th and 6th decades of life, most patients evaluated in this study belong to a +60 years age group. Although this is not conclusive and we cannot establish a statistical association, since to establish it we would need to analyze all patients with MD who consulted at the institution, a possible hypothesis is that there could be an association between the presence of DA and a +60 years age as a risk factor.  Thanks for pointing this out, we agree with this comment. Therefore, we have added this text on the page 5, paragraph 3, and line 161 as transcribed below:

“Regarding the general characteristics of the population, the average age is somewhat higher compared to that described for patients with MD (40 to 60 years), although we can find that it can occur between the third and the seventh decade of life (17). We have not found any mention in the literature about the age of DA occurrence in MS, but based on these findings, we may assume that they would be more frequent in those patients who begin with symptoms at a later age; however, and although this assertion is beyond the objectives of the present study, it is noteworthy that it opens the doors for future research in this regard.”

Comments 2: Did the author investigated for a migraine history? Other comorbidities should also me included such allergy or autoimmune diseases.

Response 2: In the collected data, we have included the history of migraine, as shown in Table 1 with a 15% prevalence, but we have not been able to associate this data with the response to betahistine with good statistical power, as was intended. Also, and as this is a study based on a review of medical records with retrospective data collection, some data regarding allergies and autoimmune diseases were not homogeneously recorded, and, therefore, we have decided not to include them in the collected data.

Comments 3: How did the authors rule out a transient stroke?

Response 3:  Drop attacks or “Tumarkin otolithic crisis” in Ménière

Here we understand falls as a sudden loss of the standing position without

compromising consciousness; and this is very important because it sets them apart from

other conditions.

They generally occur at the beginning or end of potassium paralysis caused by membrane

rupture; the patient feels as if “pushed” forward, backward or to the sides by a hand.

There is usually a pattern in each patient’s fall. Baloh recently published about a series of

elderly patients who had prior history of hearing loss (Ishiyama et al. Neurology 57:1103-

1106, 2001) which may be a specific manifestation of late MD.

Drop attacks due to a compromise in posterior circulation constitute one of the most

difficult differential diagnoses; transient ischemia of bulbar pyramids causes these falls

without loss of consciousness, generally with squat of both legs. Questioning can inquire

about the presence of other episodic symptoms during the fall or separated from it, such

as: blurred vision, double vision, hemianopsia or quadrantanopsia, alternate hemiparesis

or hemiparesthesias and migraine. A vascular history is a key factor.

Comments 4: What the vascular risk profile assessed? Cholesterol and high blood pressure may be relevant in these individuals. Please include this information in Table 1.

Response 4: These data are routinely collected from all patients, and have not been included in the Table as they are not significant.

Round 2

Reviewer 1 Report

Comments and Suggestions for Authors

Re MS    audiolres-2608146

Vestibular drop attack: an analysis of the therapeutic response  by Carmona et al.

The study evaluates the response to betahistine in patients who presented vestibular drops attacks in the context of Ménière's disease (MD) and the factors that can predict an unfa- 10 vorable response to it. This is the second evaluation.

General coments

The MS has improved and mostly follow the recommendation of reviewers. However, there are still item that should be improved, especially in the comments on limitation of the study.  Further the recommendation of shortening the conclusion has not been carried out and at present the conclusions are just repeating the discussion. There is also still linguistic problems with sometime by not using scientific terms and by using oral presentation sentences. Maybe it would help if a medical editor would correct the sentences.  Finally, the discussion has some chapters that are not involved in the studies and some items that are important in the paper are missing. Rewrite the major limitations as small number of patients in different therapeutic groups, limited time of using Betahistin, uncertainty whether outcome was based on treatment effect of betahistin or spontaneous course or more severe disease.

I will go below in detailed evaluation.

Abstract, material and methods, results

Page 1 row 16   betahistine (BTH) should not abbreviated in the abstract as the abbreviation is not used in the abstract. However in the introduction of paper the abbreviation is missing.

p1 row 16  “control is achieved in 100% of the cases.” Use better “in all cases”

p1 row 24   Suggest to use “vestibular instability”  instead of “instability disorder”

p1 row25  “an imbalance of the vestibular reflexes” prefer to use ”vestibulo-spinal reflexes”

p1 row 40  “results in inadequate otolytic stimulation”  Please correct to “otolithic”

p2 row 50  “The purpose of this study is to analyze the characteristics of the treated patients and their response to the established treatment in order to find predictors of good or bad evolution, which may be useful at the moment of deciding when to escalate in the treatment”.  What is good or bad evolution? Please specify what you measure as outcome of DA in MD

p2 row 53 “Taking into account that DA is one of the most severe manifestations of MD, and entails a great alteration in the quality of life of the individual, potentially causing serious injuries and even work disability with its economic consequences, it is vitally important to found predictors that can help us to control this dangerous symptomatology more quickly in order to avoid greater harm to the patient.” In present form it is too difficult to comprehend. Please rewrite and insert references.

p2 row 61  “An analytical, observational, cross-sectional experience was carried out, with retrospective data collection that included the review of medical records between 2015 and 2022 62 in the database of the San Lucas Foundation for Neuroscience, a neuro-otology center of 63 reference located in the city of Rosario, Argentina.  Please rewrite this sentence to scientific English. For example the patient data was collected…. Note that most of the studies are observational and analytical. Do not mention it.

p2 row 66 “these were treated with betahistine 48 mg daily (24 mg twice daily). Please avoid again repetition and shorten it to “these were treated with betahistine 24 mg twice daily.”

p2 row 66. “It must be pointed out that..” This is not written scientific Englis but oral presentation style. Please rowrite.

p2 row 79  “We have also found a considerable group of patients with unilateral hearing loss.” This sentence has no value for this paper and should be removed.

p2 row 82  “response to betahistine were recorded for all patients, those patients who were free of DA after a 1 year follow-up were considered as having an adequate response to BTH, while those patients that presented DA despite being under treatment with maximum doses of BTH (48 mg per day) during more than 3 months were considered a therapeutic failure to BTH. The treatment period did not cover the year indicating that the responses to betahistine should not evaluated. Please correct and be systematic with the abbreviation, again you use both  BTH and betahistine.  What is maximum dose of BTH? See e.g. BMJ 2016;352:h6816 http://dx.doi.org/10.1136/bmj.h6816

p3 row 120  “with a more than acceptable success rate of 100%  What is less acceptable success rate or acceptable success rate?? Please rewrite the sentence.,

p4 row 130  “It should be noted, on the one hand, that significant differences….” Please consider writing: Significant differences….

p4 row 132 “here we are referring to patients with alterations in the vestibulospinal or oculomotor (the most frequent clinical findings were nystagmus or positive head impulse tests, as well as deviation of the Bárány indices)” Note that you wrote in the material and methods that only these two tests were made and in table 1 the prevalence of pathological responses were observed in 73%???

p4 row 136  “On the other hand, response to BTH was found in 40% of the patients with less than 1 year of duration of the disease; in contrast, none of the patients in the group that did not respond to BTH had less than 1 year of onset of symptoms; although we cannot currently establish an adequate statistical association, as we do not have patients who do not respond to BTH with less than 1 year of duration of MD, this situation calls our attention.” Shorten the sentence and note that in the next sentence the reader would be confused if those patients with longer than 3 years duration of MD were not treated with BTH? Should be discussed why not.

p5 row 147. “ In this model, it can be seen with good statistical power that the presence of an abnormal physical examination predicts a lack of response to BTH (OR:7.000; CI 1.173 – 41.759; p=0.033). Likewise, the history of a disease with more than 3 years of duration strongly predicts a failure of BTH for the treatment of drop attacks due to Ménière's disease according to this model (OR:46.750; CI:4.600 – 151 475.161; p=0.001).” Use abbreviations correctly! Another explanation is that those with positive findings have more aggressive disease. Please discuss it.

p5 row 153 A multivariate logistic regression analysis was also performed between those variables that showed statistical significance in the univariate model. It is worth mentioning that only the duration of the disease for more than 3 years maintained statistical power in this analysis, with an OR:31.682; CI:2.350 – 427.066; p=0.009. The cell with more than 3 years of disease duration may be too small and therefore OR have such a great variability. Please check your statistics.

Discussion

p 5 row 169 “Drop attacks or “Tumarkin otolithic crisis” in Ménière. Here we understand falls as a sudden loss of the standing position without compromising consciousness; and this is very important because it sets them apart from other conditions.” This was stated already in introduction. please delete

p5 row 172 “They generally occur at the beginning or end of potassium paralysis caused by mem- brane rupture; the patient feels as if “pushed” forward, backward or to the sides by a hand. There is usually a pattern in each patient’s fall. Baloh recently published about a series of elderly patients who had prior history of hearing loss (21) which may be a specific manifestation of late MD. Comment: This theory is based on hypothesis and should be deleted. Reference 21 Baloh et al was not found in the literature and it was not recently.

p 5 row 172 “Drop attacks due to a compromise in posterior circulation constitute one of the most difficult differential diagnoses; transient ischemia of bulbar pyramids causes these falls without loss of consciousness, generally with squat of both legs. Questioning can inquire about the presence of other episodic symptoms during the fall or separated from it, such as: blurred vision, double vision, hemianopsia or quadrantanopsia, alternate hemiparesis or hemiparesthesias and migraine. A vascular history is a key factor.” This paragraph focuses on differential diagnosis and is partly repeating the material and methods part. It is important but should focus on the present paper.

p6 row 205 “that are so frequent in general medical and specialized neurological consultations, as it is the case with episodic vertigo. Also, we should not forget to always ask these patients about hearing loss. These two points would allow us to reach an earlier diagnosis, which is of vital importance, taking into account that the time of evolution of the disease, according to the results previously mentioned, seems to be one of the factors that determine the pharmacological response to BTH, at least in terms of DAs. It is therefore worth redoubling our efforts to reach a diagnosis as early as possible. Please correct.  This part of the chapter does not focus on the purpose of the paper and should be deleted.

p6 row 213 - 223 “Although in this study we focused on the response to DA treatment in MD and have been able to demonstrate at least a differential response when drug treatment is early versus late and when there is a normal versus abnormal physical examination, we suspect that these conclusions could be extended to patients with MD with a similar severity of rotatory attacks but without DA. However, it would be necessary to perform a similar experiment involving patients with these characteristics to prove it. There should be a discussion instead of positive effect on confounding effects as betahistin use was more than 3 months which is a short time for follow up of 1 year. Also it may be that the responses of baetahistin may be not relevant and the varaiability was caused by variability of DA may be dependent on MD severity. This discussion does not provide explanations. Please rewrite. Use abbreviations correctly.

p5 row 239 Although the mechanism of action of BTH is not completely understood, it is believed that it may exert its action as histamine H1 receptor agonist and H3 antagonist, which exerts its beneficial effects on MS at two levels, 1) increasing blood flow at the vestibular level through an improvement in the microcirculation of the stria vascularis in the cochlea, and 2) improving the central compensation of vestibular imbalances through the stimulation of histamine synthesis in the vestibular nuclei. Since MD is a pathology of the inner ear, characterized by fluctuating hydrops, it may be possible that the evolution of the pathology itself could produce structural changes in it which affect the capacity of the stria vascularis to achieve better vascularization in the inner ear in response to BTH, and thus decrease endo-lymphatic pressure. In the same way, these structural changes that would occur over time, would reveal alterations in the vestibulo-spinal or oculomotor examination that could be objectified in a possible consultation, regardless of whether or not the patient is experiencing a crisis at that time. The mechanism of Betahistin is outside of the scope of the preset paper and should be deleted.

p7 row 257. “Although in the present study we have reviewed the literature that analyzes the response to IT gentamicin in patients with MD, we have not found studies that analyze the 258 answer to this treatment in patients that present DA secondary to MD. There are at least teo papers responding the effects of gentamicin to DA. https://doi.org/10.1080/00016489850183359

Conclusions:

It is much too long and full of repetitions. Please shorten it to 10-20 rows.

p 8 row 292  As limitations of the present study we can mention a minor one…” Please rewrite the major limitations as small number of patients in different therapeutic groups, limited time of using Betahistin, uncertainty whether outcome was based on treatment effect of betahistin or spontaneous course or more severe disease.

p 8 row 293 “all patients with MS-….” Please define the abbreviation MS or replace it to MD

Comments on the Quality of English Language

There is also still linguistic problems with sometime by not using scientific terms and by using oral presentation sentences. Maybe it would help if a medical editor would correct the sentences. Maybe the Spanish papers use these oral forms of writing scientific text.

Author Response

For research article

Response to Reviewer 1 Comments

1. Summary

2. Questions for General Evaluation

Reviewer’s Evaluation

Response and Revisions

Does the introduction provide sufficient background and include all relevant references?

Must be improved

corresponding response in the point-by-point response letter

Are all the cited references relevant to the research?

Can be improved

corresponding response in the point-by-point response letter

Is the research design appropriate?

Can be improved

corresponding response in the point-by-point response letter

Are the methods adequately described?

Can be improved

corresponding response in the point-by-point response letter

Are the results clearly presented?

Can be improved

corresponding response in the point-by-point response letter

Are the conclusions supported by the results?

Must be improved

corresponding response in the point-by-point response letter

3. Point-by-point response to Comments and Suggestions for Authors

Comments 1: Page 1 row 16   betahistine (BTH) should not abbreviated in the abstract as the abbreviation is not used in the abstract. However in the introduction of paper the abbreviation is missing.

Response 1: Thanks for pointing this out. Agreed. Therefore, we have changed the abbreviation and we have included it in the introduction. You may now find the changes in the manuscript on page 2, paragraph 2 , and line 54 as transcribed below.

“For the treatment of DA (as well as for the control of MD) we have hygienic-dietary measures such as salt restriction, pharmacological measures such as betahistine (BTH) and vestibular sedatives”

Comments 2:  p1 row 16  “control is achieved in 100% of the cases.” Use better “in all cases”

Response 2: Thanks for pointing this out. Agreed. Therefore, and as recommended, wording has been modified. You may now find the changes in the manuscript on page 1, paragraph 1, and line 14 as transcribed below.

“A statistical analysis was carried out, and we found that the disease evolution time and specific alterations in the vestibulospinal and oculomotor physical examination present an unfavorable response to betahistine, as opposed to the use of intratympanic gentamicin, with which symptomatic control is achieved in all cases.”

Comments 3: p1 row 24   Suggest to use “vestibular instability”  instead of “instability disorder”

Response 3: Thanks for pointing this out. Agreed. Therefore, and as recommended, wording has been modified. Changes may be found in the manuscript on page 1, paragraph 2, and line 23 as transcribed below.

“Its etiology can be explained by vertebrobasilar insufficiency, epilepsy or vestibular instability, (the latter are also known as Tumarkin’s otolithic crises”

Comments 4: p1 row25  “an imbalance of the vestibular reflexes” prefer to use ”vestibulo-spinal reflexes”

Response 4: Thanks for pointing this out. Agreed. Therefore, and as recommended, wording has been modified. Changes may be found in the manuscript on page 1, paragraph 2, and line 25 as transcribed below.

“and their physiopathology includes the sudden imbalance of the vestibulo-spinal reflexes induced, for instance, by pathologies such as Ménière's disease (MD) in which fluctuations in vestibular tone are typical”

Comments 5: p1 row 40  “results in inadequate otolytic stimulation”  Please correct to “otolithic”

Response 5: Thanks for pointing this out. Agreed. Therefore, and as recommended, wording has been modified. Changes may be found in the manuscript on page 2, paragraph 1, and line 47 as transcribed below.

“Although the pathophysiological mechanism that produces DAs is not yet completely clear, it is believed that it could be due to a sudden change in endolymphatic flow, which may result in inadequate otolithic stimulation”

Comments 6: p2 row 50  “The purpose of this study is to analyze the characteristics of the treated patients and their response to the established treatment in order to find predictors of good or bad evolution, which may be useful at the moment of deciding when to escalate in the treatment”.  What is good or bad evolution? Please specify what you measure as outcome of DA in MD

Response 6: Thanks for pointing this out. Agreed. Therefore, we have modified wording to clarify the purpose of the study; instead of predictors of good or poor response, it seems more appropriate to say predictors of good or poor therapeutic response to betahistine. The definitions of therapeutic response are included in the Materials and Methods Section.  You may find the changes in the manuscript on page 2 , paragraph 2, and line 58 as transcribed below.

“The purpose of this study is to analyze the characteristics of the treated patients and their response to the established treatment in order to find predictors of good or bad therapeutic response, which may be useful at the moment of deciding when to escalate in the treatment”

Comments 7: p2 row 53 “Taking into account that DA is one of the most severe manifestations of MD, and entails a great alteration in the quality of life of the individual, potentially causing serious injuries and even work disability with its economic consequences, it is vitally important to found predictors that can help us to control this dangerous symptomatology more quickly in order to avoid greater harm to the patient.” In present form it is too difficult to comprehend. Please rewrite and insert references.

Response 7: Agreed. Thus, we have reviewed and modified wording to make it easier to understand, and, as you recommended, we have added bibliographical references. Changes may be found in the manuscript on page 2, paragraph 2, and line 61 as transcribed below.

“Taking into account that DA is one of the most severe manifestations of MD, and that it has a great impact on the patient's quality of life, and can cause serious injuries and even incapacity to work (21), it is of vital importance to find predictors of therapeutic response that will help us to find a way to control this dangerous symptomatology more quickly.”

Comments 8: p2 row 61  “An analytical, observational, cross-sectional experience was carried out, with retrospective data collection that included the review of medical records between 2015 and 2022 62 in the database of the San Lucas Foundation for Neuroscience, a neuro-otology center of 63 reference located in the city of Rosario, Argentina.  Please rewrite this sentence to scientific English. For example the patient data was collected…. Note that most of the studies are observational and analytical. Do not mention it.

Response 8: Thanks for pointing this out. Agreed. Therefore, and as recommended, wording has been modified. Changes may be found in the manuscript on page 2, paragraph 3, and line 69 as transcribed below.

“A cross-sectional study was conducted; patient data were collected by reviewing medical records between 2015 and 2022 in the database of the "Fundación San Lucas para la Neurociencia", a referral neurootological center located in Rosario, Argentina.”

Comments 9: p2 row 66 “these were treated with betahistine 48 mg daily (24 mg twice daily). Please avoid again repetition and shorten it to “these were treated with betahistine 24 mg twice daily.”

Response 9: Thanks for pointing this out. Agreed. Therefore, and as recommended, wording has been modified. Changes may be found in the manuscript on page 2, paragraph 3, and line 74  as transcribed below.

“these were treated with BTH 24 mg twice daily”

Comments 10: p2 row 66. “It must be pointed out that..” This is not written scientific Englis but oral presentation style. Please rowrite.

Response 10: Thanks for pointing this out. Agreed. Therefore, and as recommended, wording has been modified. Changes may be found in the manuscript on page 2, paragraph 3, and line 74 as transcribed below.

“It is worth mentioning that the patient's personal data were coded in such a way that only researchers can access them.”

Comments 11: p2 row 79  “We have also found a considerable group of patients with unilateral hearing loss.” This sentence has no value for this paper and should be removed.

Response 11: Agreed. Therefore, we have removed the sentence.

Comments 12: p2 row 82  “response to betahistine were recorded for all patients, those patients who were free of DA after a 1 year follow-up were considered as having an adequate response to BTH, while those patients that presented DA despite being under treatment with maximum doses of BTH (48 mg per day) during more than 3 months were considered a therapeutic failure to BTH. The treatment period did not cover the year indicating that the responses to betahistine should not evaluated. Please correct and be systematic with the abbreviation, again you use both  BTH and betahistine.  What is maximum dose of BTH? See e.g. BMJ 2016;352:h6816 http://dx.doi.org/10.1136/bmj.h6816

Response 12: Agreed; our wording is not clear, and so we will re-write it to make it easier to understand. All patients were treated with betahistine 48 mg daily, and received a one-year follow-up. Those patients who did not present new DAs during that period were considered to have a good answer to betahistine, while those who presented new DAs during that same period, and despite being treated with betahistine for three consecutive months, were considered to be a therapeutic failure. As regards betahistine dose and its abbreviation, we have corrected wording to make it more adequate. Thanks for pointing this out. Changes may be found in the manuscript on page 2, paragraph 3, and line 86 as transcribed below.

“The time of disease evolution (time elapsed since the appearance of the first symptoms compatible with Ménière's disease, such as vertigo or hearing symptoms) and the response to BTH were recorded in all patients, all of whom were followed up for 1 year. Patients who had no DA after 1-year follow-up were considered to have an adequate response to BTH, whereas patients who had DA despite being on treatment with standard doses of BTH for at least 3 consecutive months were considered to have therapeutic failure to BTH.”

Comments 13: p3 row 120  “with a more than acceptable success rate of 100%  What is less acceptable success rate or acceptable success rate?? Please rewrite the sentence.

Response 13: Thanks for pointing this out. Agreed. Therefore, and as recommended, wording has been modified. Changes may be found in the manuscript on page 3, paragraph 3, and line 127 as transcribed below.

“with a success rate of 100%”

Comments 14: p4 row 130  “It should be noted, on the one hand, that significant differences….” Please consider writing: Significant differences….

Response 14: Thanks for pointing this out. Agreed. Therefore, and as recommended, wording has been modified. Changes may be found in the manuscript on page 4, paragraph 2, and line 136 as transcribed below.

“Significant differences”

Comments 15: p4 row 132 “here we are referring to patients with alterations in the vestibulospinal or oculomotor (the most frequent clinical findings were nystagmus or positive head impulse tests, as well as deviation of the Bárány indices)” Note that you wrote in the material and methods that only these two tests were made and in table 1 the prevalence of pathological responses were observed in 73%???

Response 15: Agreed, our wording seems a bit unclear on this point. The physical exam carried out during consultation was a full one, and not one limited only to the presence of nystagmus, head impulse test and Barany index; besides, it was performed by expert neurootologists. We will modify wording in the Materials and Methods Section to make it easier to understand. As to the finding described in Table 1, which refers to the general characteristics of the patients included in this study, 73% of all the patients (24/33) presented some alteration during the physical exam. Table 2 analyzes, among other things, the different alteration frequencies during the physical exam in those patients who responded to BTH treatment as compared to those who did not respond to it, and this resulted in a statistically significant difference of 53.3% vs. 88.9% (p=0.047). This allows us to infer that an altered physical exam in the inter-critical periods may be a predictor of a therapeutic failure to BTH. Thanks for pointing this out. Thus, we have changed wording in the Materials and Methods Section to make it easier to understand. Changes may be found in the manuscript on page 2, paragraph 3, and line 83 as transcribed below.

“Demographic data such as age and sex, symptoms such as hearing loss, vertigo, headache, unsteadiness, pulsation, in addition to the vestibular physical examination of the patient, (a complete physical examination for alterations in the vestibulospinal or oculomotor physical examination was carried out by experienced neurootologists).”

Comments 16: p4 row 136  “On the other hand, response to BTH was found in 40% of the patients with less than 1 year of duration of the disease; in contrast, none of the patients in the group that did not respond to BTH had less than 1 year of onset of symptoms; although we cannot currently establish an adequate statistical association, as we do not have patients who do not respond to BTH with less than 1 year of duration of MD, this situation calls our attention.” Shorten the sentence and note that in the next sentence the reader would be confused if those patients with longer than 3 years duration of MD were not treated with BTH? Should be discussed why not.

Response 16: Thanks for pointing this out. Agreed. Thus, we have changed wording to shorten the sentence and make the general idea easy to understand. It is worth pointing out that, as described in the Materials and Methods Section, and as represented in the patients’ flow diagram, all the patients that participated in the study were treated with a standard dose of BTH, the differences in the highlighted lines refer to the evolution time since symptoms onset and not to a different treatment for their MD. You may find the changes in the manuscript on page 5, paragraph 1, and line 142 as transcribed below.

“On the other hand, response to BTH was found in 40% of the patients with less than 1 year of disease duration; in contrast, none of the patients in the group that did not respond to BTH had less than 1 year since the onset of symptoms. Likewise, in relation to the time of disease evolution, it was found with good statistical significance (p=<0.001) that, of the patients who responded to BTH, only 26.7% had suffered from the disease for more than 3 years; in the group of patients who did not respond to BTH, 94.4% had a duration of more than 3 years.”

Comments 17: p5 row 147. “ In this model, it can be seen with good statistical power that the presence of an abnormal physical examination predicts a lack of response to BTH (OR:7.000; CI 1.173 – 41.759; p=0.033). Likewise, the history of a disease with more than 3 years of duration strongly predicts a failure of BTH for the treatment of drop attacks due to Ménière's disease according to this model (OR:46.750; CI:4.600 – 151 475.161; p=0.001).” Use abbreviations correctly! Another explanation is that those with positive findings have more aggressive disease. Please discuss it.

Response 17: Thanks for pointing this out. Agreed. Therefore, we have changed wording to emphasize this point. You may find the changes in the manuscript on page 7, paragraph 1, and line 219 as transcribed below.

“This finding may be due to the fact that the passage of time could produce structural changes in the inner ear that make it less responsive to BTH, although we cannot rule out that we are simply facing patients with a more aggressive disease, increasing the n would be useful to clear this doubt.”

Comments 18: p5 row 153 A multivariate logistic regression analysis was also performed between those variables that showed statistical significance in the univariate model. It is worth mentioning that only the duration of the disease for more than 3 years maintained statistical power in this analysis, with an OR:31.682; CI:2.350 – 427.066; p=0.009. The cell with more than 3 years of disease duration may be too small and therefore OR have such a great variability. Please check your statistics.

Response 18: Thanks for pointing this out. We have checked our statistics and it is adequate. The cell with more than 3 years of evolution includes 21 patients; the great variability in OR seems to be due to the small n, which is one of the main study limitations; you should also keep in mind that the CI does not include the 0 value, and thus, it is statistically valid.

Comments 19: p 5 row 169 “Drop attacks or “Tumarkin otolithic crisis” in Ménière. Here we understand falls as a sudden loss of the standing position without compromising consciousness; and this is very important because it sets them apart from other conditions.” This was stated already in introduction. please delete

Response 19: Agreed. Therefore, we have deleted wording, as recommended.

Comments 20: p5 row 172 “They generally occur at the beginning or end of potassium paralysis caused by mem- brane rupture; the patient feels as if “pushed” forward, backward or to the sides by a hand. There is usually a pattern in each patient’s fall. Baloh recently published about a series of elderly patients who had prior history of hearing loss (21) which may be a specific manifestation of late MD. Comment: This theory is based on hypothesis and should be deleted. Reference 21 Baloh et al was not found in the literature and it was not recently.

Response 20: Agreed. Therefore, we have deleted wording, as recommended.

Comments 21: p 5 row 172 “Drop attacks due to a compromise in posterior circulation constitute one of the most difficult differential diagnoses; transient ischemia of bulbar pyramids causes these falls without loss of consciousness, generally with squat of both legs. Questioning can inquire about the presence of other episodic symptoms during the fall or separated from it, such as: blurred vision, double vision, hemianopsia or quadrantanopsia, alternate hemiparesis or hemiparesthesias and migraine. A vascular history is a key factor.” This paragraph focuses on differential diagnosis and is partly repeating the material and methods part. It is important but should focus on the present paper.

Response 21:  As one of the reviewers requested in the previous round, we have considered adding to the paragraph in question a description of how we reached differential diagnosis in vertebrobasilar transient strokes, but, if the reviewer considers that this is not necessary, this paragraph may be deleted. However, we have moved the paragraph to the Introduction Section, as we understand this is more appropriate, on page 1, paragraph 3, and line 28 as transcribed below.

“Los ataques de caída por compromiso de la circulación posterior constituyen uno de los diagnósticos diferenciales más difíciles; La isquemia transitoria de pirámides bulbares provoca estas caídas sin pérdida de conciencia, generalmente con sentadilla de ambas piernas. El interrogatorio puede indagar sobre la presencia de otros síntomas episódicos durante la caída o separados de ella, tales como: visión borrosa, visión doble, hemianopsia o cuadrantanopsia, hemiparesias o hemiparestesias alternas y migraña. Un historial vascular es un factor clave”.

Comentarios 22: p6 fila 205 “que son tan frecuentes en las consultas de medicina general y neurológica especializada, como es el caso de los vértigo episódicos. Además, no debemos olvidarnos de preguntar siempre a estos pacientes sobre la pérdida auditiva. Estos dos puntos nos permitirían llegar a un diagnóstico más precoz, lo cual es de vital importancia, teniendo en cuenta que el tiempo de evolución de la enfermedad, según los resultados comentados anteriormente, parece ser uno de los factores que determinan la respuesta farmacológica a BTH, al menos en términos de DA. Por tanto, vale la pena redoblar nuestros esfuerzos para llegar a un diagnóstico lo antes posible. Por favor corrija. Esta parte del capítulo no se centra en el propósito del artículo y debería eliminarse.

Respuesta 22 : De acuerdo. Por lo tanto, hemos eliminado el texto, según lo recomendado.

Comentarios 23: p6 fila 213 - 223 “Aunque en este estudio nos centramos en la respuesta al tratamiento con DA en la DM y hemos podido demostrar al menos una respuesta diferencial cuando el tratamiento farmacológico es temprano versus tardío y cuando hay un estado físico normal versus anormal examen, sospechamos que estas conclusiones podrían extenderse a pacientes con MD con una gravedad similar de ataques rotatorios pero sin DA. Sin embargo, sería necesario realizar un experimento similar con pacientes de estas características para comprobarlo. Debería haber una discusión sobre el efecto positivo sobre los efectos de confusión, ya que el uso de betahistina fue de más de tres meses, lo cual es un período corto para un seguimiento de 1 año. También puede ser que las respuestas de la baetahistina no sean relevantes y que la variabilidad causada por la variabilidad de la DA pueda depender de la gravedad de la DM. Esta discusión no proporciona explicaciones. Por favor reescribe. Utilice abreviaturas correctas.

Respuesta 23:    Si bien en este estudio nos centramos en la respuesta terapéutica de los episodios de DA en pacientes con diagnóstico de DM, hemos encontrado una asociación estadísticamente significativa en relación a pacientes con diagnóstico tardío, con una evolución de la enfermedad superior a 3 años. , o que presenten alteraciones en los exámenes físicos vestibuloespinales u oculomotores durante los períodos intercríticos. En esos pacientes, hemos encontrado que el control de la DA no se logra con una dosis estándar de BTH de 48 mg por día. Una hipótesis que surge de este estudio es que estos hallazgos pueden aplicarse al control de las crisis de vértigo en pacientes con DM, y no solo a los AD. Para ello sería necesario un análisis exhaustivo de las características de los pacientes con DM y de su respuesta terapéutica.

Esto puede deberse a que hay casos en los que la DM es más agresiva que en otros y/o a que la evolución natural de la enfermedad puede determinar que, en algunos momentos, los pacientes dejen de responder a la BTH y, por tanto, Puede ser necesario recurrir a otras opciones terapéuticas más agresivas. 

En este estudio, todos los pacientes recibieron un seguimiento de al menos 1 año. Se consideró fracaso terapéutico a aquellos pacientes que continuaron con DA a pesar de ser tratados con una dosis estándar de BTH durante 3 meses consecutivos.

Hemos modificado el texto en la Sección de Materiales y Métodos para que sea más fácil de entender, en la página 2, párrafo 3 y línea 86 como se transcribe a continuación.

“En todos los pacientes se registró el tiempo de evolución de la enfermedad (tiempo transcurrido desde la aparición de los primeros síntomas compatibles con la enfermedad de Ménière, como vértigo o síntomas auditivos) y la respuesta a la BTH, todos los cuales fueron seguidos durante al menos 1 año. . Se consideró que los pacientes que no tenían DA después de 1 año de seguimiento tenían una respuesta adecuada a la BTH, mientras que los pacientes que tenían DA a pesar de estar en tratamiento con dosis estándar de BTH durante al menos 3 meses consecutivos se consideraban que tenían fracaso terapéutico a la BTH. .”

Comentarios 24: p5 fila 239 Aunque el mecanismo de acción de la BTH no se comprende completamente, se cree que puede ejercer su acción como agonista del receptor H1 de histamina y antagonista H3, lo que ejerce sus efectos beneficiosos sobre la EM a dos niveles: 1) aumentando el flujo sanguíneo a nivel vestibular mediante una mejora de la microcirculación de la estría vascular en la cóclea, y 2) mejora de la compensación central de los desequilibrios vestibulares mediante la estimulación de la síntesis de histamina en los núcleos vestibulares. Dado que la DM es una patología del oído interno, caracterizada por hidropesía fluctuante, es posible que la propia evolución de la patología produzca cambios estructurales en la misma que afecten la capacidad de la estría vascular para lograr una mejor vascularización en el oído interno como respuesta. a BTH, y así disminuir la presión endolinfática. De la misma manera, estos cambios estructurales que se producirían con el tiempo, revelarían alteraciones en el examen vestíbulo-espinal u oculomotor que podrían objetivarse en una posible consulta, independientemente de que el paciente esté o no en crisis en ese momento. El mecanismo de Betahistin está fuera del alcance del documento preestablecido y debe eliminarse.

Respuesta 24: De acuerdo. Por lo tanto, hemos eliminado el texto, según lo recomendado.

Comentarios 25: p7 fila 257. “Aunque en el presente estudio hemos revisado la literatura que analiza la respuesta a gentamicina IT en pacientes con DM, no hemos encontrado estudios que analicen la respuesta a este tratamiento en pacientes que presentan DA secundaria a MARYLAND. Hay al menos dos artículos que responden a los efectos de la gentamicina sobre la DA. https://doi.org/10.1080/00016489850183359

Respuesta 25: Aunque en el presente estudio hemos revisado la literatura que analiza la respuesta a la gentamicina IT en pacientes con DM, hemos encontrado que la literatura que analiza la respuesta a este tratamiento en pacientes con AD en el contexto de la DM es muy escasa. . Incluso los ensayos controlados aleatorios que analizan la respuesta a la gentamicina intratimpánica en las crisis de vértigo en la DM también son escasos, y la mayoría de ellos reclutaron un número pequeño de participantes. Además, hasta la fecha, no existe acuerdo sobre los beneficios y riesgos de este tratamiento ( https://pubmed.ncbi.nlm.nih.gov/36847592/ ). Gracias por señalar esto. Acordado. Por ello, y tal como se recomienda, se ha modificado la redacción. Los cambios se pueden encontrar en el manuscrito en la página 7, párrafo 7 y línea 257 como se transcribe a continuación.

"Aunque en el presente estudio hemos revisado la literatura que analiza la respuesta a la gentamicina IT en pacientes con MD, hemos encontrado pocos estudios que analicen la respuesta a este tratamiento en pacientes que presentan DA secundaria a MD (22, 23)".

Comentarios 26: Conclusiones: Es demasiado largo y lleno de repeticiones. Recórtalo a 10-20 filas.

Respuesta 26: De acuerdo, hemos modificado la redacción como se sugirió.

Comentarios 27: p 8 fila 292 Como limitaciones del presente estudio podemos mencionar una menor…” Por favor, vuelva a escribir las limitaciones principales como un número pequeño de pacientes en diferentes grupos terapéuticos, tiempo limitado de uso de Betahistina, incertidumbre sobre si el resultado se basó en el efecto del tratamiento. de betahistina o curso espontáneo o enfermedad más grave.

Respuesta 27: Gracias por señalar esto. Acordado. Por ello, y tal como se recomienda, se ha modificado la redacción. Los cambios se pueden encontrar en el manuscrito en la página 7, párrafo 8 y línea 263, como se transcribe a continuación.

“La principal limitación de este estudio es que no es un ensayo aleatorizado, doble ciego y controlado con placebo, que sería el método ideal para construir evidencia científica, pero, aparte de eso, hay que señalar que se trata de un estudio retrospectivo. estudio con un método estadístico sólido. Además, otra limitación que debemos mencionar es el reducido número de participantes, debido a que estamos ante una complicación de una enfermedad rara; y que, para evitar errores de diagnóstico, sólo hemos incluido AD graves, es decir, aquellos que provocaron una caída al suelo. Finalmente, no podemos afirmar que un tratamiento precoz con BTH pueda garantizar que los pacientes no presenten episodios de DA, ya que es posible que, durante el curso natural de la EM, los pacientes ya no respondan a BTH; para ello sería necesario un estudio con un diseño diferente”.

Reviewer 2 Report

Comments and Suggestions for Authors

The authors have improved the paper considerably. However, there are still a few points that should be clarified. 

Comment 2: The authors write: “Although in the present study we have reviewed the literature that analyzes the response to IT gentamicin in patients with MD, we have not found studies that analyze the answer to this treatment in patients that present DA secondary to MD.” 

There is at least one case series with intratympanic steroids and gentamycin (PMID: 28033296).

Comment 13: Ménière's disease is caused by a triad consisting of vertigo episodes, fluctuating hearing and tinnitus. Though most of our patients meet the criteria for definitive MD, some of them presented the clinical triad but this did not translate into hearing loss in audiological studies, or their audiometries presented the typical ascending curve but without vertigo episodes. However, these patients meet the Bárány Society criteria for definitive MD.

Please have a second look at the diagnostic criteria of MD published by the Bárány Society. For a diagnosis of definite MD at least 2 episodes of spontaneous vertigo lasting 20 minutes to 12 hours and an audiometrically documented sensorineural hearing are required, besides fluctuating aural symptoms. Table 1 suggests that at least 18% of patients did not fulfill these criteria. 

Comment 16: The authors write “…response to BTH was found in 40% of the patients with less than 1 year of evolution of the disease; in contrast, none of the patients in the group that did not respond to BTH had less than 1 year of onset of symtoms…”

This statement still sounds contradictory. Maybe I misunderstand you but these seem to be the facts: 40 % of patients with a history of MD of less than 1 year responded to Betahistine. That means that 60 % of these did not respond, right? This contradicts the second part of the sentence where you write that none of the patients in the group that did not respond to BTH had less than 1 year of onset of symptoms. 

Comment 18: the main limitation of the study is that it is retrospective and not a randomised blinded and placebo-controlled trial. 

Author Response

Response to Reviewer 2 Comments:

1. Summary

2. Questions for General Evaluation

Reviewer’s Evaluation

Response and Revisions

Does the introduction provide sufficient back ground and include all relevant references?

Yes

corresponding response in the point-by-point response letter

Are all the cited references relevant to the research?

Can be improved

corresponding response in the point-by-point response letter

Is the research design appropriate?

Yes

corresponding response in the point-by-point response letter

Are the methods adequately described?

Yes

corresponding response in the point-by-point response letter

Are the results clearly presented?

Can be improved

corresponding response in the point-by-point response letter

Are the conclusions supported by the results?

Yes

corresponding response in the point-by-point response letter

3. Point-by-point response to Comments and Suggestions for Authors

Comment 2: The authors write: “Although in the present study we have reviewed the literature that analyzes the response to IT gentamicin in patients with MD, we have not found studies that analyze the answer to this treatment in patients that present DA secondary to MD.”

Thereis at least one case series with intratympanic steroids and gentamycin (PMID: 28033296).

Response 2: Although in the present study we have reviewed the literature that analyzes the response to IT gentamicin in patients with MD, we have found that the literature that analyzes the response to this treatment in patients with DAs in the context of MD is very scarce. Even randomized controlled trials that analyze the response to intratympanic gentamicin in vertigo crisis in MD are also scarce, and most of them recruited a small number of participants. Besides, as to date, there is no agreement as to the benefits and risks of this treatment (https://pubmed.ncbi.nlm.nih.gov/36847592/). Thanks for pointing this out. Agreed. Therefore, and as recommended, wording has been modified. Changes may be found in the manuscript on page 7, paragraph 7, and line 257 as transcribed below.

“Although in the present study we have reviewed the literature analyzing the response to IT gentamicin in patients with MD, we have found few studies analyzing the response to this treatment in patients presenting with DA secondary to MD (22, 23).”

Comment 13: Ménière's disease is caused by a triad consisting of vertigo episodes, fluctuating hearing and tinnitus. Though most of our patients meet the criteria for definitive MD, some of them presented the clinical triad but this did not translate into hearing loss in audiological studies, or their audiometries presented the typical ascending curve but without vertigo episodes. However, these patients meet the Bárány Society criteria for definitive MD.

Please have a second look at the diagnostic criteria of MD published by the Bárány Society. For a diagnosis of definite MD at least 2 episodes of spontaneous vertigo lasting 20 minutes to 12 hours and an audiometrically documented sensorineural hearing are required, besides fluctuating aural symptoms. Table 1 suggests that at least 18% of patients did not fulfill these criteria.

Response 13: Thanks for pointing this out; we agree with this comment, we have reviewed the data and corrected the wording to make it more appropriate. Some of our patients presented probable MD criteria, pursuant to the Bàrány Society criteria, though we can affirm that their DAs are of vestibular origin.

Comment 16: The authors write “…response to BTH was found in 40% of the patients with less than 1 year of evolution of the disease; in contrast, none of the patients in the group that did not respond to BTH had less than 1 year of onset of symtoms…”

This statement still sounds contradictory. Maybe I misunderstand you but these seem to be the facts: 40 % of patients with a history of MD of less than 1 year responded to Betahistine. That means that 60 % of these did not respond, right? This contradicts the second part of the sentence where you write that none of the patients in the group that did not respond to BTH had less than 1 year of onset of symptoms.

Response 16: Agreed, our wording is not clear, we will re-write it to make it easier to understand. Thanks for pointing this out. Changes may be found in the manuscript on page 5, paragraph 1, and line 142 as transcribed below.

“On the other hand, we found that, in the group of patients that presented a good response to BTH, 40 % of them had less than 1 year of symptom duration, while none of the patients in the group that did not respond to BTH had less than 1 year of symptom duration.  Likewise, in relation to evolution time of the disease, we found that, with good statistical significance (p=<0.001), only 26.7% of the patients who responded to BTH had a more than 3-year evolution of the disease, while in the group of patients who did not respond to BTH, 94.4% had a more than 3-year evolution of the disease.

Comment 18: the main limitation of the study is that it is retrospective and not a randomized blinded and placebo-controlled trial.

Response 18: Thanks for pointing this out. Agreed. Therefore, and as recommended, wording has been modified. Changes may be found in the manuscript on page 7, paragraph 8, and line 263 as transcribed below.

“The main limitation of this study is that it is not a randomized, double-blind, placebo-controlled trial, which would be the ideal method to build scientific evidence, but, apart from that, it must be pointed out that this is a retrospective study with a solid statistical method. Also, another limitation we must mention is the small number of participants, due to the fact that we are dealing with a complication of a rare disease; and that, in order to avoid diagnostic mistakes we have only included severe DA, i.e., those that resulted in a fall to the ground. Finally, we cannot affirm that an early treatment with BTH may guarantee that patients will not present DA episodes, as it is possible that, during the natural course of MD, patients may no longer respond to BTH; for this, a study with a different design would be necessary.

Round 3

Reviewer 1 Report

Comments and Suggestions for Authors

The MS has improved and mostly follow the recommendation of reviewers. However, there are still item that should be improved. 

 In the abstract the new addition claims “as opposed to the use of intratympanic gentamicin, with which symptomatic control is achieved in all cases.” The reader will get wrong impression that in this paper there would be comparison between betahistine and gentamicin. This should be corrected by stating for example “Failures for betahistine were treated with gentamicin ….. In the text  (p2 row 60) there is still “to find predictors of good or bad therapeutic response,” What is good and what is bad?,

 row 61 the new addition “Taking into account that DA is one of the most severe manifestations of MD, and that it has a great impact on the patient's quality of life, and can cause serious injuries and even incapacity to work (21), it is of vital importance to find predictors of therapeutic response that will help us to find a way to control this dangerous symptomatology more quickly.” is repetition and may be placed to the conclusions of the study.  

 row 85 “(a complete physical examination for alterations in the vestibulospinal or oculomotor physical examination was carried out by experienced neurootologists).” Does the author mean: a complete clinical evaluation of the vestibulospinal or oculomotor reflexes was carried out by experienced neurootologists?

 r 96 word ”asimmetrical”. please correct to English

 r 105 words “the lack of” is meaningless when statistical values are searched.

 Table 1 r 134. What does “Average Age (±ED) “, “Sex, M n° (%)” mean? SD  and Men? please use gender instead of sex

 The text from r 140 onward the percentages are again shown with decimals as number of subjects are only 33. Please use systematically non-decimal numbers in percentage. The same holds true also for table 2.

 r153 – 155 Again statement “the lack of response”. Please use words “efficacy” or “response” instead. Also

 r166-168 “Regarding the general characteristics of the population, the average age is somewhat higher compared to that described for patients with MD (40 to 60 years), although we canfind that it can occur between the third and the seventh decade of life (17).” Something is missing in this sentence.

 r170-173 “we may assume that they would be more frequent in those patients who begin with symptoms at a later age; however, and although this assertion is beyond the objectives of the present study, it is noteworthy that it opens the doors for future research in this regard”. This is pure speculation and should be deleted or grounded with a reference.

r180-191 “One of the main issues that this experience reveals is the great importance of making an early MD diagnosis, though it is well known that it is an under-diagnosed disease, and for which there is a significant delay in diagnosis, maybe because it is a rare disease which requires sub-specialists. This situation is even more serious if we take into account the fact that, apparently and consistently with the results of our multivariable logistic regression model, there is a moment beyond the three years in which pharmacological response is already very poor. In contrast, before the year of symptoms duration, therapeutic response to BTH is very good. This situation emphasized the importance of maximizing the efforts to reach an early diagnosis, always taking MD into account among the differential diagnoses when we care for patients with vertigo, dizziness, instability or

fluctuating drive, especially if these are accompanied by hearing symptoms or loss of cochlear function.” this paragraph is not grounded by other reports on MD and may just indicate of great variability of the symptoms of MD in longer course”. This paper focuses on DA and the paragraph is outside of the aims of the study. Could be deleted.

 r210-212  This study was designed to evaluate factors that may predict a poor initial response 211 to BTH in patients treated for drop attacks of vestibular origin secondary to Ménière's 212 disease.”. The study evaluated effect of BTH not poor effect:  Further use abbreviations systematically “drop attacks# should be abbreviates as DA.

r219-222. “This finding may be due to the fact that the passage of time could produce structural changes in the inner ear that make it less responsive to BTH, although we cannot rule out that we are simply facing patients with a more aggressive disease, increasing the n would be useful to clear this doubt.” You could add that your findings of objective test support this observation.

 r 234  Use abbreviations systemically  “drop attack”

r 239-251  Although the mechanism of action of BTH is not completely understood, it is believed that it may exert its action as histamine H1 receptor agonist and H3 antagonist, which exerts its beneficial effects on MS at two levels, 1) increasing blood flow at the vestibular level through an improvement in the microcirculation of the stria vascularis in the cochlea, and 2) improving the central compensation of vestibular imbalances through the stimulation of histamine synthesis in the vestibular nuclei. Since MD is a pathology of the inner ear, characterized by fluctuating hydrops, it may be possible that the evolution of the pathology itself could produce structural changes in it which affect the capacity of the stria vascularis to achieve better vascularization in the inner ear in response to BTH, and thus decrease endo-lymphatic pressure. In the same way, these structural changes that would occur over time, would reveal alterations in the vestibulo-spinal or oculomotor examination that could be objectified in a possible consultation, regardless of whether or not the patient is experiencing a crisis at that time.” This study did not study mechanism of action of BTH and therefore this paragraph is nonsense here. Should be deleted.  

 r 254-255  A noteworthy fact is that DA control was achieved in 100% of the patients treated with IT gentamicin, either with one or more applications”. Some comments on why the efficacy of gentamicin is 100%. This may not be true in longer course of MD. Some special injection techniques?

 The conclusions should be totally rewritten.  There are still wordings as “bad response to BTH” that may indicate poor response or severe adverse effects. The conclusions should focus on major findings and limitations of the study The statistical work up was not at all “exhaustive”, but just usual.  

Comments on the Quality of English Language

There are still unscientific wordings and errors in spelling that should be corrected

Author Response

Response to Reviewer 1 Comments

1. Summary

Thank you very much for taking the time to review this manuscript. Please find the detailed responses below and the corresponding revisions highlighted changes in the re-submitted files

2. Questions for General Evaluation

Reviewer’s Evaluation

Response and Revisions

Does the introduction provide sufficient background and include all relevant references?

Must be improved

[Please give your response if necessary. Or you can also give your corresponding response in the point-by-point response letter. The same as below]

Are all the cited references relevant to the research?

Yes

Is the research design appropriate?

Can be improved

Are the methods adequately described?

Can be improved

Are the results clearly presented?

Can be improved

Are the conclusions supported by the results?

Must be improved

3. Point-by-point response to Comments and Suggestions for Authors

Comments 1:  In the abstract the new addition claims “as opposed to the use of intratympanic gentamicin, with which symptomatic control is achieved in all cases.” The reader will get wrong impression that in this paper there would be comparison between betahistine and gentamicin. This should be corrected by stating for example “Failures for betahistine were treated with gentamicin…

Response 1: Thanks for pointing this out. Agreed. Therefore, and as recommended, wording has been modified. You may now find the changes in the manuscript on page 1, paragraph 1, and line 16 as transcribed below.

“failures for betahistine were treated with intratympanic gentamicin, with which symptomatic control is achieved in all cases”

Comments 2: In the text  (p2 row 60) there is still “to find predictors of good or bad therapeutic response,” What is good and what is bad?

Response 2: Thanks for pointing this out. Agreed. Therefore, we have changed the wording to make the text easier for the reader to understand. The terms “good” and “bad” have been replaced by “adequate” and “inadequate”, and their definitions have been included in the Materials and Methods section. You may now find this in the manuscript on page 2, paragraph 1, and line 86 as transcribed below.

“Patients who had no DA after 1-year follow-up were considered to have an adequate response to BTH, whereas patients who had DA despite being on treatment with standard doses of BTH for at least 3 consecutive months were considered to have therapeutic failure to BTH”

Comments 3: row 61 the new addition “Taking into account that DA is one of the most severe manifestations of MD, and that it has a great impact on the patient's quality of life, and can cause serious injuries and even incapacity to work (21), it is of vital importance to find predictors of therapeutic response that will help us find a way to control this dangerous symptomatology more quickly.” is repetition and may be placed to the conclusions of the study.

Response 3: Thanks for pointing this out. Agreed. Therefore, we have made the corresponding changes. You may now find the changes in the manuscript on page 7, paragraph 1, and line 238 as transcribed below.

“Taking into account that DA is one of the most severe manifestations of MD, and that it has a great impact on the patient's quality of life, and can cause serious injuries and even incapacity to work (21), it is of vital importance to find predictors of therapeutic response that will help us find a way to control this dangerous symptomatology more quickly.”

Comments 4: row 85 “(a complete physical examination for alterations in the vestibulospinal or oculomotor physical examination was carried out by experienced neurootologists).” Does the author mean: a complete clinical evaluation of the vestibulospinal or oculomotor reflexes was carried out by experienced neurootologists?

Response 4: Agreed. Therefore, we have changed the working pursuant to your recommendations. You may now find the changes in the manuscript on page 2, paragraph 3, and line 82 as transcribed below.

“(a complete clinical evaluation of the vestibulospinal or oculomotor reflexes was carried out by experienced neurootologists).”

Comments 5:  r 96 word ”asimmetrical”. please correct to English

Response 5: Thanks for pointing this out. Agreed. Therefore, we have made the corresponding changes. You may now find the changes in the manuscript on page 2, paragraph 3, and line  92 as transcribed below.

“Asymmetrical”

Comments 6: r 105 words “the lack of” is meaningless when statistical values are searched.

Response 6: Agreed. Therefore, we have changed the working pursuant to your recommendations. You may now find the changes in the manuscript on page 3, paragraph 1, and line 101 as transcribed below.

“a multivariate logistic regression model was performed to express the strength of association between the pharmacological response and the variables studied. The OR (odds ratio) and the CI were estimated at 95%.In all hypothesis contrasts, a significance level of p<0.05 was considered”

Comments 7:  Table 1 r 134. What does “Average Age (±ED) “, “Sex, M n° (%)” mean? SD  and Men? please use gender instead of sex

Response 7: Agreed. Therefore, we have changed the working pursuant to your recommendations. You may see the changes in Table 1.

Comments 8: The text from r 140 onward the percentages are again shown with decimals as number of subjects are only 33. Please use systematically non-decimal numbers in percentage. The same holds true also for table 2.

Response 8: Thanks for pointing this out. Agreed. Therefore, we have made the corresponding changes. You may see the changes throughout the text.

Comments 9:  r153 – 155 Again statement “the lack of response”. Please use words “efficacy” or “response” instead. Also

Response 9: Thanks for pointing this out. Agreed. Therefore, we have made the corresponding changes. You may now find the changes in the manuscript on page 5, paragraph 1, and line 149 as transcribed below.

“Table 3 shows the results of the univariate logistic regression in relation to the  response to BTH in the patients studied. In this model, it is observed with good statistical power that the presence of an abnormal physical examination predicts a decrease in the efficacy of BTH.”

Comments 10:  r166-168 “Regarding the general characteristics of the population, the average age is somewhat higher compared to that described for patients with MD (40 to 60 years), although we can find that it can occur between the third and the seventh decade of life (17).” Something is missing in this sentence.

Response 10: This paragraph is related to the previous one, and it may be deleted, if the reviewer so requests. It was added based on a comment made in a previous reviewers’ evaluation, in which it was mentioned that the average age is our case series was higher than the usual one presented in the consulted literature. 

This observation leads to the hypothesis that there may be a relationship between a higher age average and a higher DA frequency. This hypothesis is based on the data obtained from this study, taking into account that the study population is a subgroup of patients with MD. You may now find the changes in the manuscript on page 5, paragraph 3, and line 167 as transcribed below.

“As a hypothesis, DAs would be more frequent in those patients who start presenting symptoms at an older age; the articles that address this topic show ages similar to those found in our study”

 Comments 11: r170-173 “we may assume that they would be more frequent in those patients who begin with symptoms at a later age; however, and although this assertion is beyond the objectives of the present study, it is noteworthy that it opens the doors for future research in this regard”. This is pure speculation and should be deleted or grounded with a reference.

Response 11: Agreed. Therefore, we have added references pursuant to your recommendations.

Comments 12: r180-191 “One of the main issues that this experience reveals is the great importance of making an early MD diagnosis, though it is well known that it is an under-diagnosed disease, and for which there is a significant delay in diagnosis, maybe because it is a rare disease which requires sub-specialists. This situation is even more serious if we take into account the fact that, apparently and consistently with the results of our multivariable logistic regression model, there is a moment beyond the three years in which pharmacological response is already very poor. In contrast, before the year of symptoms duration, therapeutic response to BTH is very good. This situation emphasized the importance of maximizing the efforts to reach an early diagnosis, always taking MD into account among the differential diagnoses when we care for patients with vertigo, dizziness, instability or

fluctuating drive, especially if these are accompanied by hearing symptoms or loss of cochlear function.” this paragraph is not grounded by other reports on MD and may just indicate of great variability of the symptoms of MD in longer course”. This paper focuses on DA and the paragraph is outside of the aims of the study. Could be deleted.

Response 12: Agreed. Therefore, we have deleted the statement pursuant to your recommendations.

Comments 13: This study was designed to evaluate factors that may predict a poor initial response 211 to BTH in patients treated for drop attacks of vestibular origin secondary to Ménière's 212 disease.”. The study evaluated effect of BTH not poor effect:  Further use abbreviations systematically “drop attacks# should be abbreviates as DA.

Response 13: Thanks for pointing this out. Agreed. Therefore, we have made the corresponding changes. You may now find the changes in the manuscript on page 6, paragraph 3, and line 196 as transcribed below.

“ This study was designed to evaluate factors that may predict the response to BTH in patients treated for DA of vestibular origin secondary to Ménière's 212 disease”

Comments 14: “This finding may be due to the fact that the passage of time could produce structural changes in the inner ear that make it less responsive to BTH, although we cannot rule out that we are simply facing patients with a more aggressive disease, increasing the n would be useful to clear this doubt.” You could add that your findings of objective test support this observation.

Response 14: Thanks for pointing this out. Agreed. Therefore, we have made the corresponding changes. You may now find the changes in the manuscript on page 6, paragraph 3, and line 205 as transcribed below.

This finding may be due to the fact that the passage of time could produce structural changes in the inner ear that make it less responsive to BTH, although we cannot rule out that we are simply facing patients with a more aggressive disease, the findings of this study make it possible for us to support this observation, increasing the n would be useful to clear this doubt.

Comments 15:: r 239-251  “Although the mechanism of action of BTH is not completely understood, it is believed that it may exert its action as histamine H1 receptor agonist and H3 antagonist, which exerts its beneficial effects on MS at two levels, 1) increasing blood flow at the vestibular level through an improvement in the microcirculation of the stria vascularis in the cochlea, and 2) improving the central compensation of vestibular imbalances through the stimulation of histamine synthesis in the vestibular nuclei. Since MD is a pathology of the inner ear, characterized by fluctuating hydrops, it may be possible that the evolution of the pathology itself could produce structural changes in it which affect the capacity of the stria vascularis to achieve better vascularization in the inner ear in response to BTH, and thus decrease endo-lymphatic pressure. In the same way, these structural changes that would occur over time, would reveal alterations in the vestibulo-spinal or oculomotor examination that could be objectified in a possible consultation, regardless of whether the patient is experiencing a crisis at that time or not.” This study did not study mechanism of action of BTH and therefore this paragraph is nonsense here. Should be deleted.  

Response 15: Agreed. Therefore, we have deleted the statement pursuant to your recommendations.

Comments 16: r 254-255  “A noteworthy fact is that DA control was achieved in 100% of the patients treated with IT gentamicin, either with one or more applications”. Some comments on why the efficacy of gentamicin is 100%. This may not be true in longer course of MD. Some special injection techniques?

Response 16: Though it is true that we cannot affirm that, during the course of MD, patients will no present new DA episodes, based on our experience, all the patients who were treated with intratympanic gentamicin (applied 1 or up to 3 times), presented no new DA episodes. We can affirm this based on the fact that we have followed up some of our patients for more than 5 years.

The technique used is:  The patient is placed in position and 90% phenol is applied in the anterior inferior quadrant and then in the posterior inferior quadrant as a local anaesthetic. Myringotomy is performed in the anterior inferior quadrant and then 0.5 ml of gentamicin 30 mg is applied through the postero inferior quadrant in the middle ear under otomicroscopic vision.

Comments 16: The conclusions should be totally rewritten.  There are still wordings as “bad response to BTH” that may indicate poor response or severe adverse effects. The conclusions should focus on major findings and limitations of the study The statistical work up was not at all “exhaustive”, but just usual.  

Thanks for pointing this out. Agreed. Therefore, we have made the corresponding changes. You may now find the changes in the manuscript on page 7, paragraph 1, and line 238 as transcribed below.

Conclusions:

Taking into account that DA is one of the most severe manifestations of MD, and that it has a great impact on the patient's quality of life, and can cause serious injuries and even incapacity to work (21), it is of vital importance to find predictors of therapeutic response that will help us to find a way to control this dangerous symptomatology more quickly.

The present study supports us, to state that in patients who present severe DA secondary to MD, the response to pharmacological treatment with BTH is conditioned, on the one hand, by the duration of symptoms, being the efficacy very good in patients with less than a year of MD duration, and poor when the MD have been present for more than three years. On the other hand, another strong predictive factor of a bad response to BTH is the presence of an impaired vestibulo-spinal or oculomotor physical examination in intercritical periods, although, to be rigorous, we may only sustain this last statement when it is carried out by people with expertise in this matter.

The conclusions we were able to obtain open the door to different questions. Bearing in mind that we have found strong predictors of a poor response to BTH, such as abnormal vestibular physical examination and a disease with an evolution time of more than 3 years, is it possible to directly consider treatment with intratympanic gentamicin in those patients who present all these characteristics? In case a treatment with BTH is attempted in these patients, for how long do we apply it before considering it a therapeutic failure? (8, 15). 

It would also be necessary to carry out a similar experience with an exhaustive statistical analysis in all the patients with MD in order to determine if the relationships found at the time of controlling the vestibular DA have the same implications for the control of the symptoms and the evolution of the disease, which would expand the number of patients who would possibly benefit from these findings.

Reviewer 2 Report

Comments and Suggestions for Authors

.

Author Response

Thank you for taking the time to review this manuscript. Your input has been very valuable in order to improve the quality of our work. Attached is the latest version of the manuscript according to the suggestions of the other reviewers. 
We take this opportunity to send you our best regards and look forward to a favorable response.